# How to build a consistency model:
# Learning flow maps via self-distillation

**Nicholas M. Boffi**
Carnegie Mellon University

**Michael S. Albergo**
Harvard University

**Eric Vanden-Eijnden**
Courant Institute of Mathematical Sciences

## Abstract

Flow-based generative models achieve state-of-the-art sample quality, but require the expensive solution of a differential equation at inference time. Flow map models, commonly known as consistency models, encompass many recent efforts to improve inference-time efficiency by learning the *solution operator* of this differential equation. Yet despite their promise, these models lack a unified description that clearly explains how to learn them efficiently in practice. Here, building on the methodology proposed in Boffi et al. (2024), we present a systematic algorithmic framework for directly learning the flow map associated with a flow or diffusion model. By exploiting a relationship between the velocity field underlying a continuous-time flow and the instantaneous rate of change of the flow map, we show how to convert any distillation scheme into a direct training algorithm via self-distillation, eliminating the need for pre-trained teachers. We introduce three algorithmic families based on different mathematical characterizations of the flow map: Eulerian, Lagrangian, and Progressive methods, which we show encompass and extend all known distillation and direct training schemes for consistency models. We find that the novel class of Lagrangian methods, which avoid both spatial derivatives and bootstrapping from small steps by design, achieve significantly more stable training and higher performance than more standard Eulerian and Progressive schemes. Our methodology unifies existing training schemes under a single common framework and reveals new design principles for accelerated generative modeling. Associated code is available at https://github.com/nmboffi/flow-maps.

## 1 Introduction

Generative models based on dynamical systems, such as flows and diffusions, have achieved remarkable successes in vision (Song et al., 2020; Rombach et al., 2022; Ma et al., 2024; Polyak et al., 2025), language (Lou et al., 2024), protein structure prediction (Abramson et al., 2024), weather forecasting (Price et al., 2024), and materials design (Zeni et al., 2025). While highly expressive, dynamical models leverage the solution of a differential equation for sample generation, which typically requires repeated evaluation of the learned model. This computational bottleneck has limited the application of flows and diffusions in domains where rapid inference is crucial, such as real-time control (Black et al., 2024; Chi et al., 2024) and image editing, and as a result has led to intense interest in accelerated inference. One particularly promising approach, which underlies consistency models (Song et al., 2023; Kim et al., 2024), is to estimate the *flow map* associated with the deterministic probability flow equation instead of the velocity field governing its instantaneous dynamics. The flow map is defined by the integrated flow and can be used to generate samples in as few as one model evaluation, leading to inference that can be $10 - 100\times$ faster than traditional dynamical models. This dramatic speedup

39th Conference on Neural Information Processing Systems (NeurIPS 2025).

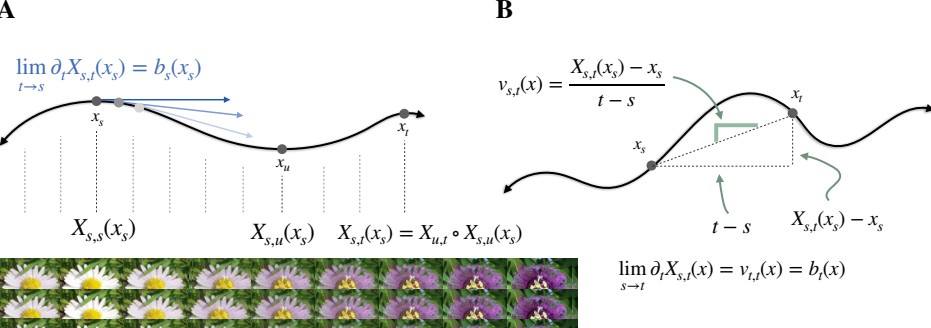

**A**

$$\lim_{t \to s} \partial_t X_{s,t}(x_s) = b_s(x_s)$$

$$X_{s,s}(x_s) \qquad X_{s,u}(x_s) \quad X_{s,t}(x_s) = X_{u,t} \circ X_{s,u}(x_s)$$

**B**

$$v_{s,t}(x) = \frac{X_{s,t}(x_s) - x_s}{t - s}$$

$$t - s \qquad X_{s,t}(x_s) - x_s$$

$$\lim_{s \to t} \partial_t X_{s,t}(x) = v_{t,t}(x) = b_t(x)$$

**Figure 1: Overview.** (A) Schematic of the two-time flow map $X_{s,t}$ and the tangent condition (Lemma 2.1), which provides a relation between the map and the drift of the probability flow. The flow map is composable, invertible, and has the property that as $t \to s$, its time derivative recovers the drift $b_s$ from (2). (B) Illustration of our proposed parameterization. The function $v_{s,t}$ estimates the slope of the line drawn between two points on a trajectory of the probability flow, and can be directly trained efficiently via the tangent condition.

potential motivates the central question: is there a principled methodology for training flow maps, and how can we do so efficiently in practice? In this work,

*We introduce a direct training framework for flow maps, eliminating the need for pre-trained teacher models while maintaining the training stability of distillation.*

Recently, there have been broad efforts to learn the flow map either directly or through distillation of a pre-trained model (Boffi et al., 2024; Frans et al., 2024; Zhou et al., 2025; Salimans et al., 2024). Distillation-based approaches perform well empirically, but require a two-phase learning setup in which the performance of the student is limited by the performance of the teacher. In these methods, the practitioner first learns a score (Song and Ermon, 2020; Ho et al., 2020) or flow (Lipman et al., 2022; Albergo and Vanden-Eijnden, 2022; Albergo et al., 2023; Liu et al., 2022) model, and then converts it into a flow map via a secondary training algorithm that considers the pre-trained model as a "teacher" for the "student" flow map. To avoid this complication, we aim to design learning schemes in which the flow map can be trained similarly to standard flow matching. In this endeavor, the fundamental challenge is a lack of a unified mathematical characterization that reveals how to learn the flow map efficiently, which has led to complex pipelines that require extensive engineering to overcome unstable optimization dynamics outside of the distillation setting (Lu and Song, 2025).

To address this challenge, we introduce a mathematical framework that exposes a landscape of novel training schemes. Our key insight is a simple relation, *the tangent condition* (Figure 1), that explicitly relates the velocity of the probability flow equation to the derivative of the flow map. Using this insight, we develop a self-distillation framework where the flow map is learned by simultaneously training and distilling its implicit velocity. The result is a simple pipeline that leverages off-the-shelf training procedures for flows to learn a model with accelerated inference. Our framework reveals the fundamental design principles for learning flow maps, enabling practitioners to build few-step generative models as systematically as standard flows. Our main contributions are:

1. **Algorithmic framework.** We provide three equivalent mathematical characterizations of the flow map, showing how consistency models and other recent few-step methods – including consistency trajectory models (Kim et al., 2024), shortcut models (Frans et al., 2024), mean flow (Geng et al., 2025), and align your flow (Sabour et al., 2025) – emerge as special cases of our methodology.

2. **Self-distillation algorithms.** Leveraging our description of the flow map, we introduce three new algorithmic families – Eulerian (ESD), Lagrangian (LSD), and Progressive Self-Distillation (PSD) – and discuss their connections to existing direct training schemes. We prove that each has the correct unique minimizer, and provide guarantees that the loss values bound the 2-Wasserstein error of the learned one-step model for ESD and LSD.

3. **Empirical analysis.** We study the performance of each method as a function of the number of spatial and time derivatives that appear in the objective function. We find that LSD, which avoids both spatial derivatives and self-consistent bootstrapping from smaller steps, attains the best performance across standard benchmarks including the synthetic checkerboard dataset, CIFAR-10, CelebA-64, and AFHQ-64.

## 2 Theoretical framework

In this work, we study the *flow map* of the probability flow equation, which is a function that jumps between points along a trajectory (Figure 1). Given access to the flow map, samples can be generated in a single step by jumping directly to the endpoint, or can be generated with an adaptive amount of computation at inference time by taking multiple steps. Below, we give a detailed mathematical description of the flow map, which we leverage to design a suite of novel training schemes. We begin with a review of stochastic interpolants, which we use to build efficient flow-based generative models.

### 2.1 Stochastic interpolants and probability flows

Let $\mathcal{D} = \{x_1^i\}_{i=1}^n$ with each $x_1^i \in \mathbb{R}^d$, $x_1^i \sim \rho_1$ denote a dataset drawn from a target density $\rho_1$. Given $\mathcal{D}$, our goal is to draw a fresh sample $\hat{x}_1 \sim \hat{\rho}_1$ from a distribution $\hat{\rho}_1 \approx \rho_1$ learned to approximate the target. Recent methods for accomplishing this task leverage flows, which dynamically evolve samples from a simple base distribution $\rho_0$ such as a Gaussian until they resemble samples from $\rho_1$.

**Interpolants.** To build a flow-based generative model, we leverage the stochastic interpolant framework (Albergo et al., 2023), which we now briefly recall. We define a *stochastic interpolant* as a stochastic process $I : [0, 1] \times \mathbb{R}^d \times \mathbb{R}^d \to \mathbb{R}^d$ that combines samples from the target and the base,

$$I_t(x_0, x_1) = \alpha_t x_0 + \beta_t x_1, \tag{1}$$

where $\alpha, \beta : [0, 1] \to [0, 1]$ are continuously differentiable functions satisfying the boundary conditions $\alpha_0 = 1, \alpha_1 = 0, \beta_0 = 0$, and $\beta_1 = 1$. In (1), the pair $(x_0, x_1) \sim \rho(x_0, x_1)$ is drawn from a coupling satisfying the marginal constraints $\int_{\mathbb{R}^d} \rho(x_0, x_1) dx_0 = \rho_1(x_1)$ and $\int_{\mathbb{R}^d} \rho(x_0, x_1) dx_1 = \rho_0(x_0)$. By construction, the probability density $\rho_t = \mathsf{Law}(I_t)$ defines a path in the space of measures between the base and the target. This path specifies a probability flow that pushes samples from $\rho_0$ onto $\rho_1$,

$$\dot{x}_t = b_t(x_t), \qquad x_0 \sim \rho_0, \tag{2}$$

which has the same distribution as the interpolant, $x_t \sim \rho_t$ for all $t \in [0, 1]$. The drift $b$ in (2) is given by the conditional expectation of the time derivative of the interpolant, $b_t(x) = \mathbb{E}[\dot{I}_t | I_t = x]$, which averages the "velocity" of all interpolant paths that cross the point $x$ at time $t$. A standard choice of coefficients is $\alpha_t = 1 - t$ and $\beta_t = t$ (Albergo and Vanden-Eijnden, 2022; Albergo et al., 2023), which recovers flow matching (Lipman et al., 2022) and rectified flow (Liu et al., 2022). Many other options have been considered in the literature, and in addition to flow matching, variance-preserving and variance-exploding diffusions can be obtained as particular cases.

**Learning.** By standard results in probability theory and statistics, the conditional expectation $b_t$ can be learned efficiently in practice by solving a square loss regression problem,

$$b = \underset{\hat{b}}{\arg\min}\, \mathcal{L}_b(\hat{b}), \qquad \mathcal{L}_b(\hat{b}) = \int_0^1 \mathbb{E}_{x_0, x_1}\big[|\hat{b}_t(I_t) - \dot{I}_t|^2\big] dt. \tag{3}$$

Above, $\mathbb{E}_{x_0, x_1}$ denotes an expectation over the random draws of $(x_0, x_1)$ in the interpolant (1).

**Sampling.** Given an estimate $\hat{b}$ obtained by minimizing (3) over a class of neural networks, we can generate an approximate sample $\hat{x}_1$ by numerically integrating the learned probability flow $\dot{\hat{x}}_t = \hat{b}_t(\hat{x}_t)$ until time $t = 1$ from an initial condition $\hat{x}_0 \sim \rho_0$. This approach yields high-quality samples from complex data distributions in practice, but is computationally expensive due to the need to repeatedly evaluate the learned model during integration; here, we aim to avoid this solve.

### 2.2 Characterizing the flow map

The *flow map* $X : [0, 1]^2 \times \mathbb{R}^d \to \mathbb{R}^d$ is the unique map satisfying the jump condition

$$X_{s,t}(x_s) = x_t \text{ for all } (s, t) \in [0, 1]^2, \tag{4}$$

where $(x_t)_{t \in [0,1]}$ is any solution of (2). The condition (4) means that the flow map takes "steps" of arbitrary size $t - s$ along trajectories of the probability flow. In particular, a single application $X_{0,1}(x_0)$

with $x_0 \sim \rho_0$ yields a sample from $\rho_1$, avoiding numerical integration entirely. Moreover, we may also increase the number of steps by composing $X_{t_i, t_{i+1}}$ over a grid $0 = t_0 < t_1 < ... < t_k = 1$ in the presence of model errors, which enables us to trade inference-time compute for sample quality.

In what follows, we give three characterizations of the flow map that each lead to an objective for its estimation. As we now show, these characterizations are based on a simple but key result that shows we can deduce the corresponding velocity field $b_t$ from a given flow map $X_{s,t}$.

**Lemma 2.1** (Tangent condition). *Let $X_{s,t}$ denote the flow map. Then,*

$$\lim_{s \to t} \partial_t X_{s,t}(x) = b_t(x) \qquad \forall t \in [0, 1], \quad \forall x \in \mathbb{R}^d, \tag{5}$$

*i.e. the tangent vectors to the curve $(X_{s,t}(x))_{t \in [s,1]}$ give the velocity field $b_t(x)$ for every $x$.*

As illustrated in Figure 1A, Lemma 2.1 highlights that there is a velocity model "implicit" in a flow map. To leverage this algorithmically, we propose to adopt an Euler step-like parameterization that takes into account the boundary condition $X_{s,s}(x) = x$,

$$X_{s,t}(x) = x + (t - s)v_{s,t}(x). \tag{6}$$

In (6), $v : [0, 1]^2 \times \mathbb{R}^d \to \mathbb{R}^d$ is the function we will estimate parametrically. Despite its similarity to a first-order Taylor expansion, the representation (6) corresponds to a shift and rescaling of $X_{s,t}$, and hence is without loss of expressivity. In addition to enforcing that $X_{s,t}$ recovers the identity on the diagonal $s = t$, (6) implies that $\lim_{s \to t} \partial_t X_{s,t}(x) = v_{t,t}(x)$, which gives an elegant connection between $v_{s,t}$ and the drift field $b_t$,

$$v_{t,t}(x) = b_t(x), \qquad \forall t \in [0, 1], \quad \forall x \in \mathbb{R}^d. \tag{7}$$

Geometrically, $v_{s,t}$ describes the "slope" of the line drawn between $x_s$ and $x_t$ on a single ODE trajectory (Figure 1B). The condition (7) states that the slope between two infinitesimally-spaced points is precisely the velocity $b_t$. A key insight is that this relation indicates $v_{t,t}$ can be estimated using the objective (3). To learn the map $X_{s,t}$, it then remains to estimate $v_{s,t}$ away from the diagonal $s = t$. To this end, we leverage the following result, which relates $v_{s,t}$ to $v_{t,t}$ for $s \neq t$.

**Proposition 2.2** (Flow map). *Assume that $X_{s,t}$ is given by (6) with $v_{s,t}$ satisfying (7), and assume that $v_{s,t}$ is continuous in both time arguments. Then, $X_{s,t}$ is the flow map defined in (4) if and only if any of the following conditions also holds:*

*(i) (Lagrangian condition): $X_{s,t}$ solves the Lagrangian equation*

$$\partial_t X_{s,t}(x) = v_{t,t}(X_{s,t}(x)), \tag{8}$$

*for all $(s, t) \in [0, 1]^2$ and for all $x \in \mathbb{R}^d$.*

*(ii) (Eulerian condition): $X_{s,t}$ solves the Eulerian equation*

$$\partial_s X_{s,t}(x) + \nabla X_{s,t}(x) v_{s,s}(x) = 0, \tag{9}$$

*for all $(s, t) \in [0, 1]^2$ and for all $x \in \mathbb{R}^d$.*

*(iii) (Semigroup condition): For all $(s, t, u) \in [0, 1]^3$ and for all $x \in \mathbb{R}^d$,*

$$X_{u,t}(X_{s,u}(x)) = X_{s,t}(x). \tag{10}$$

The Lagrangian and Eulerian conditions in Proposition 2.2 categorize the flow map $X_{s,t}$ as the solution of an infinite system of ODEs or as the solution of a PDE, each of which describes transport along trajectories of the flow (2). The semigroup condition states that any two jumps can be replaced by a single jump. Sections B to D provide a review of the flow map matching framework (Boffi et al., 2024), and describe how these three characterizations are the basis for consistency (Song et al., 2023; Kim et al., 2024; Geng et al., 2024) and progressive distillation (Salimans and Ho, 2022a) schemes that have appeared in the literature. In the following, we show how each – and in fact how *any* distillation method that produces a flow map from a velocity field $\hat{b}$ – can be immediately converted into a *direct training* objective for a single network model $X_{s,t}$ via the concept of self-distillation.

## 2.3 A framework for self-distillation

Our framework augments training $v_{t,t}$ on the diagonal $s = t$ via the objective (3) and the identity (7) with a penalization term for one or more of the conditions in Proposition 2.2 along the off-diagonal $s \neq t$. This leads to a set of objectives that can each be used to learn the flow map.

**Proposition 2.3** (Self-distillation). *The flow map $X_{s,t}$ defined in (4) is given for all $0 \leqslant s \leqslant t \leqslant 1$ by $X_{s,t}(x) = x + (t - s)v_{s,t}(x)$ where $v_{s,t}(x)$ the unique minimizer over $\hat{v}$ of*

$$\mathcal{L}_{\mathsf{SD}}(\hat{v}) = \mathcal{L}_b(\hat{v}) + \mathcal{L}_{\mathsf{D}}(\hat{v}), \tag{11}$$

*where $\mathcal{L}_b(\hat{v})$ is given by*

$$\mathcal{L}_b(\hat{v}) = \int_0^1 \mathbb{E}_{x_0, x_1} \big[ |\hat{v}_{t,t}(I_t) - \dot{I}_t|^2 \big] dt, \tag{12}$$

*and where $\mathcal{L}_{\mathsf{D}}(\hat{v})$ is any of the following three objectives.*

*(i) The* Lagrangian self-distillation (LSD) *objective, which leverages (8),*

$$\mathcal{L}_{\mathsf{LSD}}(\hat{v}) = \int_0^1 \int_0^t \mathbb{E}_{x_0, x_1} \big[ |\partial_t \hat{X}_{s,t}(I_s) - \hat{v}_{t,t}(\hat{X}_{s,t}(I_s))|^2 \big] ds dt; \tag{13}$$

*(ii) The* Eulerian self-distillation (ESD) *objective, which leverages (9),*

$$\mathcal{L}_{\mathsf{ESD}}(\hat{v}) = \int_0^1 \int_0^t \mathbb{E}_{x_0, x_1} \big[ |\partial_s \hat{X}_{s,t}(I_s) + \nabla \hat{X}_{s,t}(I_s) \hat{v}_{s,s}(I_s)|^2 \big] ds dt; \tag{14}$$

*(iii) The* progressive self-distillation (PSD) *objective, which leverages (10),*

$$\mathcal{L}_{\mathsf{PSD}}(\hat{v}) = \int_0^1 \int_0^t \int_s^t \mathbb{E}_{x_0, x_1} \big[ |\hat{X}_{s,t}(I_s) - \hat{X}_{u,t}(\hat{X}_{s,u}(I_s))|^2 \big] du ds dt. \tag{15}$$

*Above, $\hat{X}_{s,t}(x) = x + (t - s)\hat{v}_{s,t}(x)$ and $\mathbb{E}_{x_0, x_1}$ denotes an expectation over the random draws of $(x_0, x_1)$ in the interpolant defined in (1).*

The proof follows directly from Proposition 2.2 and is given in Section E; the resulting algorithmic approach is summarized graphically in Figure 2. In Proposition 2.3, we focus on $s \leqslant t$, which is all that is required to generate data. Training over the entire unit square $(s, t) \in [0, 1]^2$ enables jumping backwards from data to noise along trajectories of (2) in addition to standard generation from noise to data. The derivatives with respect to space and time required to implement the LSD and ESD losses can be computed efficiently via standard `jvp` implementations. As we will see in our experiments, an advantage of the schemes (13) and (15) is they naturally avoid derivatives with respect to space, which leads to significantly improved training stability.

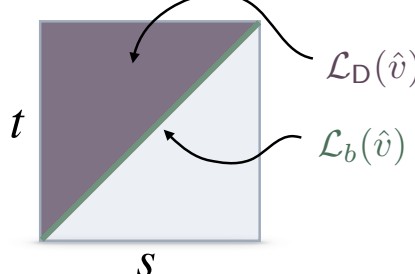

$$\mathcal{L}_{\mathsf{SD}}(\hat{v}) = \mathcal{L}_b(\hat{v}) + \mathcal{L}_{\mathsf{D}}(\hat{v})$$

**Figure 2: Self-distillation.** Our plug-and-play approach pairs any distillation objective $\mathcal{L}_{\mathsf{D}}$ on the off-diagonal $s \neq t$ of the square $[0, 1]^2$ with a flow matching objective $\mathcal{L}_b$ on the diagonal $s = t$ to obtain a direct training algorithm for the flow map.

We now provide theoretical guarantees that the objective value bounds the accuracy of the model for LSD and ESD. We were unable to obtain a similar guarantee for PSD due to issues of compounding errors and distribution shift associated with bootstrapping small steps to large steps, which we believe to be a fundamental difficulty to the algorithm's construction. This difficulty is consistent with the observed reduced performance of PSD in comparison to LSD in our experiments (Section 5).

**Proposition 2.4** (Wasserstein bounds). *Let $\hat{X}_{s,t}(x) = x + (t - s)\hat{v}_{s,t}(x)$ denote a candidate flow map, let $\hat{\rho}_1 = \hat{X}_{0,1} \sharp \rho_0$ denote the corresponding one-step generated distribution, and let $\hat{L}$ denote the spatial Lipschitz constant of $\hat{v}_{t,t}(\cdot)$ uniformly in $t$. First assume $\mathcal{L}_b(\hat{v}) + \mathcal{L}_{\mathsf{LSD}}(\hat{v}) \leqslant \varepsilon$. Then,*

$$\mathsf{W}_2^2(\hat{\rho}_1, \rho_1) \leqslant 4e^{1+2\hat{L}}\varepsilon. \tag{16}$$

*Now assume that $\mathcal{L}_b(\hat{v}) + \mathcal{L}_{\mathsf{ESD}}(\hat{v}) \leqslant \varepsilon$. Then,*

$$\mathsf{W}_2^2(\hat{\rho}_1, \rho_1) \leqslant 2e \cdot (1 + e^{2\hat{L}})\varepsilon. \tag{17}$$

---

**Algorithm 1:** Learning flow maps via self-distillation

---

**input:** Dataset $\mathcal{D}$; interpolant coefficients $\alpha_t$, $\beta_t$; batch size $M$; diagonal fraction $\eta$; distillation
      method $\mathcal{L}_{\mathsf{D}} \in \{\mathcal{L}_{\mathsf{LSD}}, \mathcal{L}_{\mathsf{ESD}}, \mathcal{L}_{\mathsf{PSD}}\}$.

**repeat**

> Sample $M_d = \lfloor \eta M \rfloor$ pairs $(x_0^i, x_1^i) \sim \rho(x_0, x_1)$ and times $t_i \sim U([0, 1])$;
> Compute interpolants $I_{t_i} = \alpha_{t_i} x_0^i + \beta_{t_i} x_1^i$ and velocities $\dot{I}_{t_i} = \dot{\alpha}_{t_i} x_0^i + \dot{\beta}_{t_i} x_1^i$;
> Compute diagonal loss: $\mathcal{L}_b = \frac{1}{M_d} \sum_{i=1}^{M_d} e^{-w_{t_i, t_i}} |\hat{v}_{t_i, t_i}(I_{t_i}) - \dot{I}_{t_i}|^2 + w_{t_i, t_i}$;
> Sample $M_o = M - M_d$ pairs $(x_0^j, x_1^j) \sim \rho(x_0, x_1)$ and times $(s_j, t_j) \sim U_{\mathsf{od}}$;
> Compute interpolants $I_{s_j} = \alpha_{s_j} x_0^j + \beta_{s_j} x_1^j$;
> Compute distillation loss: $\mathcal{L}_{\mathsf{D}} = \frac{1}{M_o} \sum_{j=1}^{M_o} e^{-w_{s_j, t_j}} \mathcal{L}_{\mathsf{D}}^{s_j, t_j}(\hat{v}) + w_{s_j, t_j}$;
> Compute $\mathcal{L}_{\mathsf{SD}} = \mathcal{L}_b + \mathcal{L}_{\mathsf{D}}$;
> Update $\hat{v}$ and $w$ using $\nabla \mathcal{L}_{\mathsf{SD}}$;

**until** *converged*;

**output:** Trained flow map $\hat{X}_{s,t}(x) = x + (t - s)\hat{v}_{s,t}(x)$

---

The above result highlights that the model's accuracy improves systematically as the loss is minimized for both ESD and LSD. The proof follows by combining guarantees for distillation-based algorithms (Boffi et al., 2024) with guarantees for flow-based algorithms (Albergo et al., 2023), which can be stitched together by our assumption on the value of the loss.

## 3 Algorithmic aspects

We now provide practical numerical recommendations for an implementation of self-distillation. Our aim is not to provide a single best method, but to devise a general-purpose framework that can be used to build high-performing flow maps across data modalities. We provide a general algorithmic prescription in Algorithm 1, with specific instantiations for LSD, ESD, and PSD in Section F.6.

**Choice of teacher.** The self-distillation objectives in Proposition 2.3 are obtained by squaring the residuals of the properties in Proposition 2.2. While the minimizers are correct, the associated training dynamics may not be optimal because the losses are nonconvex in $\hat{v}$. The flow of information should be from the diagonal $\hat{v}_{t,t}$, where there is an external learning signal via $\dot{I}_t$, to the off-diagonal $\hat{v}_{s,t}$, which bootstraps the signal in $\hat{v}_{t,t}$. To enforce that $\hat{v}_{s,t}$ learns to match $\hat{v}_{t,t}$, rather than vice-versa, we use the stopgrad operator to match the distillation setting, where the off-diagonal would adapt entirely to an external teacher. Detailed descriptions of the recommended placement are given in Section F.4.

**Relation to existing methods.** The generic framework described by Proposition 2.3 and Algorithm 1 recovers most existing schemes for training consistency models and their extensions. In particular, by proper choice of the distillation objective and the teacher, we can obtain standard training for consistency models (Song et al., 2023), consistency trajectory models (Kim et al., 2024), and shortcut models (Frans et al., 2024). These connections are given in detail in Sections C and D.

**Loss weighting.** The loss (11) can be written explicitly as an integral over the upper triangle $s < t$,

$$\mathcal{L}_{\mathsf{SD}}(\hat{v}) = \int_0^1 \int_0^t \left( \mathcal{L}_b^t(\hat{v}) \delta(s - t) + \mathcal{L}_{\mathsf{D}}^{s,t}(\hat{v}) \right) ds\, dt, \tag{18}$$

where $\mathcal{L}_b^t(\hat{v}) = \mathbb{E}_{x_0, x_1}[|\hat{v}_{t,t}(I_t) - \dot{I}_t|^2]$ denotes (12) restricted to $t$, and where $\mathcal{L}_{\mathsf{D}}^{s,t}(\hat{v})$ denotes the distillation term restricted $(s, t)$. We find that loss values at different pairs $(s, t)$ can have gradient norms that differ significantly, introducing undesirable variance. To rectify this, we incorporate a learned weight $w_{s,t}$, generalizing the EDM2 weight (Karras et al., 2024) to the two-time setting,

$$\mathcal{L}_{\mathsf{SD}}(\hat{v}) = \int_0^1 \int_0^t \left( e^{-w_{s,t}} \left( \mathcal{L}_b^t(\hat{v}) \delta(s - t) + \mathcal{L}_{\mathsf{D}}^{s,t}(\hat{v}) \right) + w_{s,t} \right) ds\, dt. \tag{19}$$

In (19), $w_{s,t}$ can be interpreted as an estimate of the log-variance of the loss values; at the global minimizer, it ensures that all values of $(s, t)$ contribute on a similar scale. We find that using $w_{s,t}$ significantly stabilizes the training dynamics and enables the use of larger learning rates.

**Temporal sampling.**    In addition to the weight, we introduce a sampling distribution $p_{s,t}$,

$$\mathcal{L}_{\mathsf{SD}}(\hat{v}) = \mathbb{E}_{p_{s,t}} \left[ e^{-w_{s,t}} \left( \mathcal{L}_b^t(\hat{v})\delta(s-t) + \mathcal{L}_{\mathsf{D}}^{s,t}(\hat{v}) \right) + w_{s,t} \right]. \tag{20}$$

While the weight $w_{s,t}$ normalizes the variance across times, $p_{s,t}$ chooses how we select times randomly for each batch. Let $U_{\mathsf{d}}$ denote the uniform distribution on the diagonal $s = t$, and let $U_{\mathsf{od}}$ denote the uniform distribution on the upper triangle $s < t$. In our experiments, we leverage the mixture distribution $p_{s,t} = \eta U_{\mathsf{d}} + (1 - \eta)U_{\mathsf{od}}$, which places a fraction $\eta$ of the batch uniformly at random on the diagonal and a fraction $(1 - \eta)$ uniformly at random in the upper triangle. Because our distillation losses reduce to the flow matching loss in the limit as $s \to t$ (Section F.2), we only use $\mathcal{L}_b$ on the diagonal $s = t$. We found $\eta = 0.75$ to work well in early experiments, which puts the majority of the computational effort towards learning the flow and proportionally less towards distilling it. The distillation objectives in Proposition 2.3 are more expensive than the interpolant objective $\mathcal{L}_b$ because they require multiple evaluations of the network and Jacobian-vector products. As a result, $\eta$ can also be used to tune the cost of each training step (Section F.3).

**PSD sampling.**    For PSD, we introduce a proposal distribution $p_u$ over the intermediate step,

$$\mathcal{L}_{\mathsf{PSD}}^{s,t}(\hat{v}) = \mathbb{E}_{p_u}\mathbb{E}_{x_0,x_1}\left[ \left| \hat{X}_{s,t}(I_s) - \hat{X}_{u,t}(\hat{X}_{s,u}(I_s)) \right|^2 \right]. \tag{21}$$

We parameterize $u = \gamma s + (1 - \gamma)t$ as a convex combination for $\gamma \in [0, 1]$ and define the proposal distribution by sampling over $\gamma$. In our experiments, we compare uniform sampling (PSD-U) with $\gamma \sim U([0, 1])$ to midpoint sampling where $\gamma \sim \delta_{1/2}$ so that $\gamma = 1/2$ deterministically (PSD-M).

**PSD scaling.**    We show in Section F that (21) may be rewritten entirely in terms of $\hat{v}$ as

$$\mathcal{L}_{\mathsf{PSD}}^{s,t}(\hat{v}) = (t-s)^2 \mathbb{E}_{p_\gamma}\mathbb{E}_{x_0,x_1}\left[ \left| \hat{v}_{s,t}(I_s) - (1-\gamma)\hat{v}_{s,u}(I_s) - \gamma\hat{v}_{u,t}(\hat{X}_{s,u}(I_s)) \right|^2 \right]. \tag{22}$$

The form (22) eliminates factors $u - s$, $t - s$, and $t - u$ that appear due to the parameterization (6). We found these terms introduced higher gradient variance because they cause the loss to scale like $(t - s)^2$, which changes the effective learning rate depending on the timestep $(t - s)$; we drop this factor of $(t - s)^2$ in practice. (22) preconditions the loss and removes this additional source of variability, leading to improved training stability.

**Conditioning and guidance.**    Flow maps can be made conditional by incorporating a conditioning argument $c$ as $X_{s,t}(x; c) = x + (t - s)v_{s,t}(x; c)$. We can use this observation to define classifier-free guided (CFG) flow maps, as we now show (Ho and Salimans, 2022). To do so, let $c = \varnothing$ correspond to unconditional generation, and let $q_t(x; \alpha, c) = b_t(x; \varnothing) + \alpha \left( b_t(x; c) - b_t(x; \varnothing) \right)$ be the CFG velocity field at guidance strength $\alpha$. This velocity has a flow map $X_{s,t}(x; \alpha, c) = x + (t - s)v_{s,t}(x; \alpha, c)$ satisfying $v_{t,t}(x; \alpha, c) = q_t(x; \alpha, c)$, which may be learned via self-distillation by incorporating additional random sampling over the guidance scale (Section F.5). In this work, we focus solely on unconditional, unguided generation and leave the usage of guidance for future study.

**Multiple models and general representations.**    In Proposition 2.3, we leverage a single model $\hat{X}_{s,t}$ defined in terms of $\hat{v}_{s,t}$. While this leads to greater efficiency through (7), it requires one model to learn both the velocity $b$ and its flow map $X$, which may require more network capacity than traditional flow-based generative models. Instead, it is also possible to use two models – one parameterizing $b$ and one parameterizing $X$ – which can be trained simultaneously. Higher-order parameterizations can be designed that leverage both, such as $\hat{X}_{s,t}(x) = x + (t - s)\hat{b}_s(x) + \frac{1}{2}(t - s)^2\hat{\psi}_{s,t}(x)$, which can use a frozen pre-trained model $\hat{b}_s$ or can train $\hat{b}$ from scratch in tandem with $\hat{\psi}$. More generally, any parameterization $\hat{X}_{s,t}$ satisfying $\hat{X}_{s,s}(x) = x$ may be used in practice, where in this setting we use $\lim_{s \to t} \partial_t \hat{X}_{s,t}(x)$ in place of $\hat{v}_{t,t}(x)$ in Algorithm 1. This may be computed via automatic differentiation as a `jvp` in $t$ at $s = t$. In this work, we focus on the representation (6), which requires only a single model and gives a computationally efficient way to evaluate $\mathcal{L}_b$; we leave these more general and higher-order parameterizations to future work.

## 4    Related work

**Flow matching and diffusion models.**    Our approach builds directly on methods from flow matching and stochastic interpolants (Lipman et al., 2022; Albergo and Vanden-Eijnden, 2022; Albergo

| Dataset | Method | Step Count | | | | |
|---|---|---|---|---|---|---|
| | | 1 | 2 | 4 | 8 | 16 |
| Checker (KL ↓) | LSD | **0.086** | **0.077** | **0.071** | **0.070** | 0.071 |
| | ESD | 0.098 | 0.092 | 0.083 | 0.082 | 0.075 |
| | PSD-M | 0.146 | 0.089 | 0.081 | 0.072 | 0.069 |
| | PSD-U | 0.111 | 0.107 | 0.075 | 0.073 | **0.068** |
| CIFAR-10 (FID ↓) | LSD | **8.100** | **4.370** | **3.340** | **3.330** | **3.570** |
| | PSD-M | 12.810 | 8.430 | 5.960 | 5.070 | 4.640 |
| | PSD-U | 13.610 | 7.950 | 6.030 | 5.320 | 5.160 |
| CelebA-64 (FID ↓) | LSD | **12.220** | **5.740** | **3.180** | **2.180** | **1.960** |
| | PSD-M | 19.640 | 11.750 | 7.890 | 6.060 | 5.090 |
| | PSD-U | 18.810 | 11.020 | 7.470 | 6.000 | 5.630 |
| AFHQ-64 (FID ↓) | LSD | **11.190** | **7.780** | **7.000** | **5.890** | **5.610** |
| | PSD-M | 18.860 | 14.750 | 14.400 | 13.260 | 11.070 |
| | PSD-U | 14.500 | 10.730 | 10.990 | 12.020 | 11.470 |

Table 1: **Benchmark results.** Performance across sampling step counts for the low-dimensional checker dataset (KL divergence) and natural image datasets (FID). Best method per dataset and step count shown in **bold**.

et al., 2023; Liu et al., 2022) as well as the probability flow equation associated with diffusion models (Song et al., 2020; Ho et al., 2020; Maoutsa et al., 2020; Boffi and Vanden-Eijnden, 2023). These methods define an ordinary differential equation whose solution evaluates the flow map at a single time. Due to the computational expense associated with solving these equations, a line of recent work asks how to resolve the flow more efficiently with higher-order numerical solvers (Dockhorn et al., 2022; Lu et al., 2022; Karras et al., 2022; Li et al., 2024) and parallel sampling schemes (Chen et al., 2024; Bortoli et al., 2025). Our approach instead estimates the flow map to enable accelerated sampling by avoiding the differential equation solve altogether.

**Consistency models.** Appearing under several names, the flow map has become a central object of study in recent efforts to obtain accelerated inference. Consistency models (Song et al., 2023; Song and Dhariwal, 2023) estimate the single-time flow map to jump from any time $s$ to data, given by $X_{s,1}$ in our notation. Consistency trajectory models (Kim et al., 2024; Li and He, 2025; Luo et al., 2023) estimate the two-time flow map, enabling multistep sampling. Both approaches implicitly leverage the Eulerian characterization (9), which we find leads to gradient instability, explaining recent engineering efforts for stable training (Lu and Song, 2024). Progressive distillation (Salimans and Ho, 2022a) uses the semigroup condition (10) to train a model that can recursively replicate two steps of a pre-trained teacher. Progressive flow map matching (Boffi et al., 2024) enforces this iteratively over a flow map after pre-training, while shortcut models apply a discretized semigroup condition (Frans et al., 2024). In concurrent work, Geng et al. (2025); Sabour et al. (2025) introduce distillation and direct training schemes that reduce to a particular case of our Eulerian formulation. Details on these methods and their connection to our framework may be found in Sections B to D.

## 5 Numerical experiments

We test LSD, ESD, PSD-U, and PSD-M on the low-dimensional checkerboard dataset, as well as in the high-dimensional setting of unconditional image generation on CIFAR-10, CelebA-64, and AFHQ-64. In each case, we study performance at fixed training time to obtain a fair comparison. We emphasize that our aim is not to obtain state of the art performance, but to understand the trade-offs of each approach and compare them on an equal footing; with further engineering, quantitative metrics could be lowered significantly for all methods. For image datasets, we find ESD to be unstable due to the spatial gradient, leading to poor performance without gradient stabilization schemes. We find that LSD obtains uniformly the best performance on all problems tried. This is consistent with our theoretical results in Proposition 2.4, where we were able to obtain stronger theoretical guarantees for LSD than for PSD. Full network and training details are provided in Section G and Table 2.

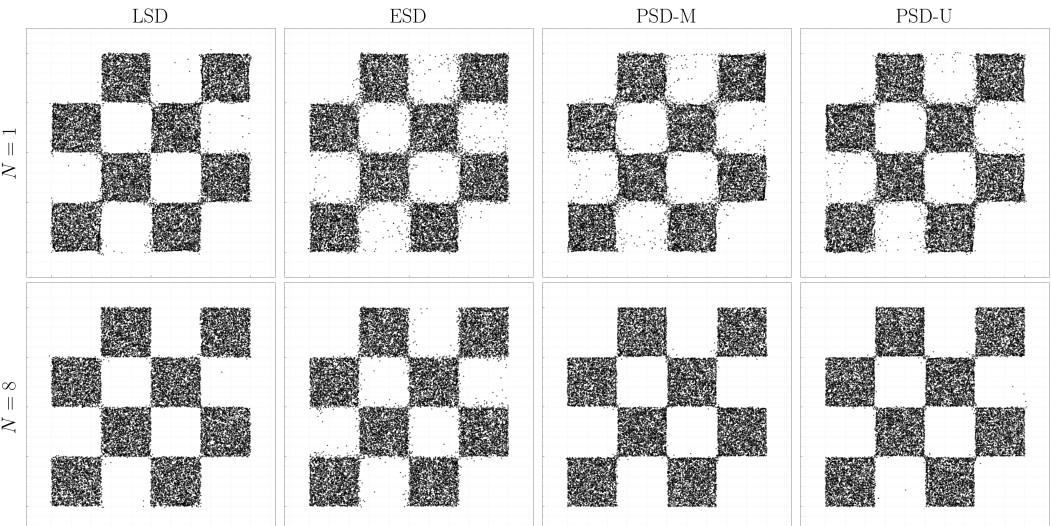

**Figure 3: Checker dataset.** Qualitative results for the two-dimensional checker dataset. LSD performs the best across all step counts except $N = 16$ (Table 1). All methods improve as the number of steps increase. ESD and both PSD variants fail to capture the sharp boundaries at small $N$, introducing artifacts and driving KL higher.

**Checkerboard.** While synthetic, the checkerboard dataset exhibits multimodality, sharp boundaries, and low-dimensionality that make it a useful testbed for exact visualization of how few-step samplers capture complex features in the target. Qualitative results are shown in Figure 3, while quantitative results obtained by estimating the KL divergence (for details, again see Section G) between generated samples and the target are shown in Table 1. LSD performs best across all sampling steps tried except for $N = 16$, where all methods perform well. The performance of LSD also saturates around $N = 4$ sampling steps. By contrast, ESD, PSD-U, and PSD-M all see increased performance up to $N = 16$ steps with reduced performance for fewer steps. The qualitative results in Figures 3 and 6 highlight that the higher KL values result from a failure to capture the sharp features present in the dataset, with ESD blurring the boundaries and the PSD methods introducing artifacts that connect the modes.

**CIFAR-10.** In Figure 4, we study the parameter gradient norm as a function of the training iteration on CIFAR-10. LSD and PSD, which avoid computing spatial derivatives of the network during training, maintain significantly more stable gradients than ESD even when using $\mathsf{sg}\,(\cdot)$. We found the high gradient norm of ESD to induce training instability, ultimately leading to divergence. This is consistent with earlier work on consistency models, where careful annealing schedules, clipping, and network design has been necessary to stabilize continuous-time training (Lu and Song, 2025).

We track the quantitative performance of each method as measured by FID in Table 1; we do not report FID values for ESD due to training instability. We find that LSD obtains the best performance across all step counts followed by PSD. PSD-U and PSD-M trade places depending on step count. A qualitative visualization of sample quality is shown in Figure 5 (Top) as a function of the number of sampling steps. We see that each method obtains improved quality as the number of steps increases, and that all methods produce similar images for fixed seed.

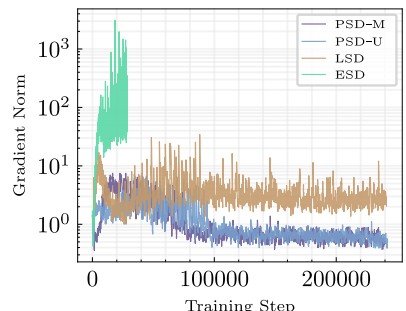

**Figure 4: CIFAR-10: Parameter gradient norms.** Spatial and temporal representations in the flow map impact parameter gradient norms of self-distillation methods that require network time and space derivatives.

**CelebA-64.** As shown in Table 1, LSD also obtains the best performance across all step counts on CelebA-64 (Liu et al., 2015), with FID scores ranging from 12.22 at $N = 1$ to 1.96 at $N = 16$. The gap between LSD and the PSD variants is more pronounced on CelebA-64 than on CIFAR-10, particularly for low step counts. PSD-U mostly outperforms PSD-M, with PSD-M only obtaining a higher-

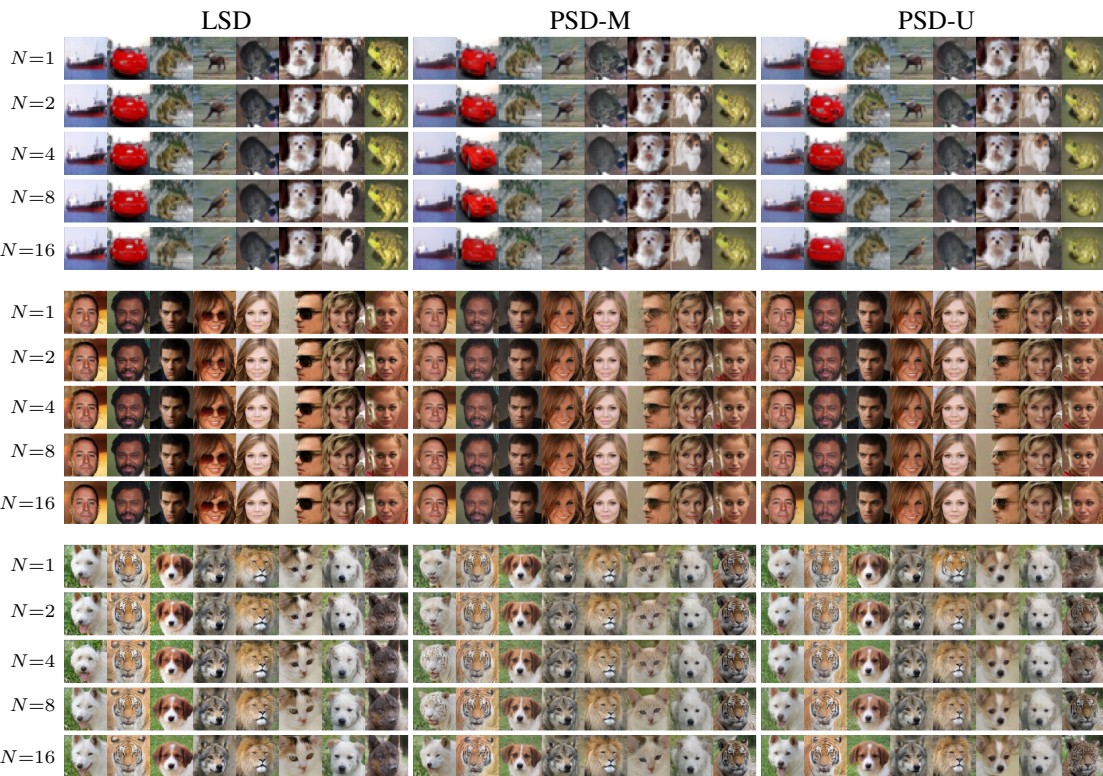

**Figure 5: Progressive refinement.** Sample quality as a function of sampling steps using the same eight fixed noise samples across all methods for fair comparison. **(Top)** CIFAR-10, **(Middle)** CelebA-64, **(Bottom)** AFHQ-64. LSD consistently produces coherent samples across all datasets and step counts.

performing 16-step map. A qualitative visualization is shown in Figure 5 (Middle). All methods show systematic improvement as the step count increases, with faces becoming sharper and more detailed.

**AFHQ-64.** Finally, we evaluate on AFHQ-64, a more challenging dataset with greater visual diversity than CelebA-64 that includes variation across animal categories (Choi et al., 2020). As shown in Table 1, LSD again achieves the best FID scores across all step counts, ranging from 11.19 at $N = 1$ to 5.61 at $N = 16$. PSD shows notably higher FID scores on this dataset, particularly PSD-M, which struggles at low step counts. PSD-U again mostly outperforms PSD-M, with PSD-M obtaining a slightly higher-performing 16-step map but worse performance otherwise. Qualitative results are shown in Figure 5 (Bottom), where we again see that higher step counts lead to generated images with increasing levels of detail.

## 6 Conclusion

In this work, we expose and investigate the design space of a class of flow-based generative models with accelerated inference known as *flow maps*. These models generalize and extend consistency models to include multiple training paradigms and principled multistep inference. Rather than learning the velocity field typical of flows and diffusions, flow maps learn the solution operator of the probability flow equation, obviating the need to solve a differential equation for inference. We show that learning can be performed directly by pairing flow training with any of three characterizations of the flow map, an approach we refer to as self-distillation. Self-distillation can be incorporated with minimal additional overhead, making flow maps an appealing new paradigm. While we systematically categorize the design space of flow map models, the main limitation of our contribution is that we were unable to systematically test each component empirically due to the large associated computational expense. Critical aspects deserving further experimentation include ablations over the flow map parameterization and architecture; stabilization, annealing, and stopgradient schemes for training; and hybrid approaches that combine multiple of our self-distillation objectives.

## Acknowledgments

MSA is supported by a Junior Fellowship at the Harvard Society of Fellows as well as the National Science Foundation under Cooperative Agreement PHY-2019786 (The NSF AI Institute for Artificial Intelligence and Fundamental Interactions, http://iaifi.org/). NMB would like to thank Max Simchowitz, Andrej Risteski, Stephen Huan, Jerry Huang, Chaoyi Pan, Giri Anantharaman, and Gabe Guo for helpful conversations.

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

# A    Background on stochastic interpolants

For the reader's convenience, we now recall how to construct probability flow equations using stochastic interpolants. We remark that score-based diffusion models can also be cast in the form of a stochastic interpolant (1), though they do not usually satisfy the exact boundary conditions at $t = 0$ and $t = 1$, and the direction of time is opposite by convention. For example, the variance exploding and EDM processes (Karras et al., 2022) naturally fit within this form as $X_t = X_0 + \sigma_t z$, while the variance preserving process can be cast in this form after solving the Ornstein-Uhlenbeck process $dX_t = -X_t dt + \sqrt{2} dW_t$ in distribution as $X_t \stackrel{d}{=} \exp(-t) X_0 + \sqrt{1 - \exp(-2t)} z$ with $z \sim \mathsf{N}(0, I)$. In both cases, we may define $I_t = X_{T-t}$ to flip the direction of time over the horizon $T$.

The key property of the interpolant construction is that solutions to the probability flow (2) push forward their initial conditions onto samples from the target by matching the time-dependent density of (1), as we now show.

**Lemma A.1** (Transport equation). *Let $\rho_t = \mathsf{Law}(I_t)$ be the density of the stochastic interpolant (1). Then $\rho_t$ satisfies the transport equation*

$$\partial_t \rho_t(x) + \nabla \cdot (b_t(x)\rho_t(x)) = 0, \qquad \rho_{t=0}(x) = \rho_0(x) \tag{23}$$

*where $\nabla$ denotes a gradient with respect to $x$, and where $b_t(x) = \mathbb{E}[\dot{I}_t \mid I_t = x]$.*

*Proof.* The proof proceeds via the weak form of (23). Let $\phi \in C_b^1(\mathbb{R}^d)$ denote an arbitrary continuously differentiable and compactly supported test function. By definition,

$$\forall t \in [0, 1] \quad : \quad \int_{\mathbb{R}^d} \phi(x)\rho_t(x)dx = \mathbb{E}[\phi(I_t)] \tag{24}$$

where $I_t$ is given by (1). Taking the time derivative of this equality, we deduce that

$$\begin{aligned}
\int_{\mathbb{R}^d} \phi(x)\partial_t \rho_t(x)dx &= \mathbb{E}[\dot{I}_t \cdot \nabla \phi(I_t)] \\
&= \mathbb{E}[b_t(I_t) \cdot \nabla \phi(I_t)] \\
&= \int_{\mathbb{R}^d} b_t(x) \cdot \nabla \phi(x)\rho_t(x)dx.
\end{aligned} \tag{25}$$

The first line follows by the chain rule, the second by the tower property of the conditional expectation and the definition of the drift $b_t$, and the third by definition of $\rho_t$. The last line is the weak form of the transport equation (23). $\qquad \square$

A nearly-identical derivation (simply dropping the tower property step) shows that the probability flow equation (2) satisfies $\mathsf{Law}(x_t) = \rho_t = \mathsf{Law}(I_t)$. Together, these results imply that we can sample from any density $\rho_t$ solving a transport equation of the form (23) by solving the corresponding ordinary differential equation (2). In practice, we may implement this algorithmically by approximating $b_t$ with a neural network via minimization of (3) to obtain a model $\hat{b}_t$, and then solving the associated differential equation $\dot{\hat{x}}_t = \hat{b}_t(\hat{x}_t)$ from $t = 0$ to $t = 1$ with an initial condition $\hat{x}_0 \sim \rho_0$ to obtain approximate samples from $\rho_1$.

# B    Background on flow map matching.

As discussed in Boffi et al. (2024), given a pre-trained velocity field $\hat{b}$, we may leverage the three properties in Proposition 2.2 to design efficient distillation schemes by minimizing the corresponding square residual. In the following, we use the notation $\mathcal{L}(\hat{X}; \hat{b})$ or $\mathcal{L}(\hat{X}; \check{X})$ to denote a loss function for the flow map $\hat{X}$ given the teacher (which remains frozen during training).

**Stopgradients.** Because $\hat{b}$ is a pre-trained teacher, its parameters are frozen during training. The self-distillation schemes we introduce in this work replace the teacher network $\hat{b}_s$ by a self-consistent implicit teacher $\hat{v}_{s,s}$, eliminating the need for the pre-trained model entirely. Inspired by the distillation setting, in Section F.4, we will use a stopgradient operator $\mathsf{sg}(\cdot)$ in the context of self-distillation schemes to create a similar effect to a frozen teacher and to control the flow of information within the model. Nevertheless, for training stability, it has been observed that it can be useful to use additional $\mathsf{sg}(\cdot)$ operators even for distillation, which we discuss after introducing each loss.

## B.1 Lagrangian distillation.

The first approach is the Lagrangian map distillation (LMD) algorithm, which is based on (8) and is the basis for the LSD algorithm,

$$\mathcal{L}_{\mathsf{LMD}}(\hat{X}; \hat{b}) = \int_0^1 \int_0^t \mathbb{E}_{\rho_s} \left[ |\partial_t \hat{X}_{s,t}(I_s) - \hat{b}_t(\hat{X}_{s,t}(I_s))|^2 \right] ds dt. \tag{26}$$

The Lagrangian scheme (26) was introduced in Boffi et al. (2024), and to our knowledge has not appeared in other works. While $\hat{b}$ is frozen, the loss (26) is nonconvex in $\hat{X}$ due to the nonlinearity of $\hat{b}$. Moreover, computing the gradient of (26) with respect to $\hat{X}$ (or its parameters) requires computing the spatial Jacobian of $\hat{b}$, which has been observed to be problematic for large generative models such as image synthesis systems (Poole et al., 2022). For these reasons, it is common to use the modified loss function

$$\mathcal{L}_{\mathsf{LMD}}(\hat{X}; \hat{b}) = \int_0^1 \int_0^t \mathbb{E}_{\rho_s} \left[ |\partial_t \hat{X}_{s,t}(I_s) - \hat{b}_t \left( \mathsf{sg}\left( \hat{X}_{s,t}(I_s) \right) \right)|^2 \right] ds dt. \tag{27}$$

The effectiveness of (27) over (26) depends on the data modality and the neural network architecture, as the spatial Jacobian is only problematic in some contexts depending on the pre-trained teacher. We refer to the gradient of a loss function such as (27) – which includes the $\mathsf{sg}\,(\cdot)$ operator – as a *semigradient*.

## B.2 Eulerian distillation.

A second scheme is the Eulerian map distillation (EMD) method based on (9),

$$\mathcal{L}_{\mathsf{EMD}}(\hat{X}; \hat{b}) = \int_0^1 \int_0^t \mathbb{E}_{\rho_s} \left[ |\partial_s \hat{X}_{s,t}(I_s) + \nabla \hat{X}_{s,t}(I_s) \hat{b}_s(I_s)|^2 \right] ds dt. \tag{28}$$

Unlike the Lagrangian approach, (28) is convex in $\hat{X}$. Nevertheless, taking the gradient with respect to the parameters of $\hat{X}$ requires backpropagating through its spatial Jacobian, which can be similarly problematic as the setting described for (26). One fix is to use a semigradient based on

$$\mathcal{L}_{\mathsf{EMD}}(\hat{X}; \hat{b}) = \int_0^1 \int_0^t \mathbb{E}_{\rho_s} \left[ |\partial_s \hat{X}_{s,t}(I_s) + \mathsf{sg}\left( \nabla \hat{X}_{s,t}(I_s) \hat{b}_s(I_s) \right)|^2 \right] ds dt, \tag{29}$$

which avoids backpropagating through the spatial Jacobian entirely. While this helps training stability, it has been observed by Boffi et al. (2024) that the Lagrangian schemes (26) and (27) are more stable than the Eulerian schemes (28) and (29), which is consistent with our experiments in Section 5.

## B.3 Progressive flow map matching.

We now describe the progressive flow map matching (PFMM) algorithm, which is inspired by progressive distillation (Salimans and Ho, 2022b) for diffusion models, but adapted to the stochastic interpolant and two-time flow map setting. Let $\check{X}_{s,t}$ denote a pre-trained teacher flow map, assumed to be valid over the range $0 \leqslant s \leqslant t \leqslant \tau$. To obtain such a map at initialization, we may take $\tau = \Delta t$ and set $\check{X}_{s,t}(x) = x + (t - s)\hat{b}_s(x)$ with a pre-trained flow map $\hat{b}$, corresponding to a single Euler step of size $(t - s) \leqslant \tau = \Delta t$. Our aim is to "extend" $\check{X}$ over a larger range, say $0 \leqslant s \leqslant t \leqslant 2\tau$, by training a second flow map $\hat{X}_{s,t}$ to match two steps of $\check{X}_{s,t}$. To do so, we consider the objective

$$\mathcal{L}_{\mathsf{PFMM}}(\hat{X}; \check{X}) = \int_0^{2\tau} \int_0^t \int_s^t \mathbb{E}_{\rho_s} \left[ |\hat{X}_{s,t}(I_s) - \check{X}_{u,t}\left( \check{X}_{s,u}(I_s) \right)|^2 \right] du ds dt, \tag{30}$$

which is based on the semigroup property (10). In words, (30) teaches $\hat{X}$ to replicate two jumps of $\check{X}$ in one larger jump. We may also apply (30) self-consistently, where $\hat{X}$ itself serves as the teacher,

$$\mathcal{L}_{\mathsf{PFMM}}(\hat{X}) = \int_0^{2\tau} \int_0^t \int_s^t \mathbb{E}_{\rho_s} \left[ |\hat{X}_{s,t}(I_s) - \mathsf{sg}\left( \hat{X}_{u,t}\left( \hat{X}_{s,u}(I_s) \right) \right)|^2 \right] du ds dt, \tag{31}$$

after the first round where $\hat{b}$ is used, and extend $\tau$ over the course of optimization according to a pre-defined annealing scheme. Our general self-distillation framework described in Section 2.3 may be obtained by using one of the above distillation schemes in tandem with direct training of $\hat{v}$, and where we use $\hat{v}$ as the teacher velocity field for the student flow map model $\hat{X}$.

# C  Connection to consistency models.

The approaches (28) and (29) are directly related to consistency distillation in the continuous-time limit (Song and Dhariwal, 2023; Lu and Song, 2024). Consistency models estimate the single-time flow map from noise to data, which in our notation is given by $X_{s,1}$. Consistency trajectory models (Kim et al., 2024) use the same approach to learn the two-time map $X_{s,t}$; for agreement with the main text, we focus on this setting here.

## C.1  Consistency distillation and Align Your Flow.

We first take a continuous-time limit of the discrete-time consistency distillation objective. Discrete-time consistency distillation considers the loss

$$\mathcal{L}_{\mathsf{CD}}(\hat{X}; \hat{b}) = \int_0^1 \int_0^t \mathbb{E}\left[|\hat{X}_{s,t}(I_s) - \mathsf{sg}\left(\hat{X}_{s+\Delta s,t}(\hat{x}_{s+\Delta s})\right)|^2\right],$$

$$I_s = \alpha_s x_0 + \beta_s x_1,$$

$$\hat{x}_{s+\Delta s} = I_s + \Delta s \hat{b}_s(I_s).$$

(32)

In words, $\mathcal{L}_{\mathsf{CD}}$ aims to make $\hat{X}_{s,t}$ "consistent" on trajectories of the teacher's probability flow $\hat{x}_t$ by using a shorter step of size $(t - s - \Delta s)$ as a teacher for a slightly larger step of size $(t - s)$. Taking the gradient with respect to $\hat{X}_{s,t}$, we find

$$\frac{\delta \mathcal{L}_{\mathsf{CD}}}{\delta \hat{X}_{s,t}}(\hat{X}; \hat{b}) = \hat{X}_{s,t}(I_s) - \hat{X}_{s+\Delta s,t}(\hat{x}_{s+\Delta s}).$$

(33)

To obtain the gradient with respect to the parameters $\theta$ of $\hat{X}$, we have by the chain rule that $\nabla_\theta \mathcal{L}_{\mathsf{CD}} = \nabla_\theta \hat{X}_{s,t} \frac{\delta \mathcal{L}_{\mathsf{CD}}}{\delta \hat{X}_{s,t}}$, so we focus on the functional derivative for notational simplicity. Taylor expanding, we find that

$$
\begin{aligned}
\hat{X}_{s+\Delta s,t}(\hat{x}_{s+\Delta s}) &= \hat{X}_{s+\Delta s,t}(I_s + \Delta s \hat{b}_s(I_s)), \\
&= \hat{X}_{s+\Delta s,t}(I_s) + \Delta s \nabla \hat{X}_{s+\Delta s,t}(I_s) \hat{b}_s(I_s) + o(\Delta s), \\
&= \hat{X}_{s,t}(I_s) + \Delta s \partial_s \hat{X}_{s,t}(I_s) + \Delta s \nabla \hat{X}_{s+\Delta s,t}(I_s) \hat{b}_s(I_s) + o(\Delta s) \\
&= \hat{X}_{s,t}(I_s) + \Delta s \left(\partial_s \hat{X}_{s,t}(I_s) + \nabla \hat{X}_{s,t}(I_s) \hat{b}_s(I_s)\right) + o(\Delta s)
\end{aligned}
$$

(34)

In the last line, we used that $\Delta s \nabla \hat{X}_{s+\Delta s,t}(I_s) = \Delta s \nabla \hat{X}_{s,t}(I_s) + o(\Delta s)$. With this, we find

$$\lim_{\Delta s \to 0} \frac{1}{\Delta s} \frac{\delta \mathcal{L}_{\mathsf{CD}}}{\delta \hat{X}_{s,t}}(\hat{X}; \hat{b}) = -\left(\partial_s \hat{X}_{s,t}(I_s) + \nabla \hat{X}_{s,t}(I_s) \hat{b}_s(I_s)\right),$$

(35)

which is simply the negative Eulerian residual.

We now ask if the semigradient (35) can be obtained from the Eulerian distillation objective (28) with a certain choice of $\mathsf{sg}(\cdot)$. To do so, we consider the specific parameterization (6) given by $\hat{X}_{s,t}(x) = x + (t - s)\hat{v}_{s,t}(x)$. In this case, the Eulerian equation becomes

$$
\begin{aligned}
&\partial_s \hat{X}_{s,t}(x) + \nabla \hat{X}_{s,t}(x) \hat{b}_s(x) \\
&= -\hat{v}_{s,t}(x) + (t - s)\partial_s \hat{v}_{s,t}(x) + \hat{b}_s(x) + (t - s)\nabla \hat{v}_{s,t}(x)\hat{b}_s(x).
\end{aligned}
$$

(36)

As a result, the Eulerian map distillation loss (28) becomes

$$\mathcal{L}_{\mathsf{EMD}}(\hat{v}; \hat{b})$$
$$= \int_0^1 \int_0^t \mathbb{E}_{\rho_s}\left[|-\hat{v}_{s,t}(I_s) + (t - s)\partial_s \hat{v}_{s,t}(I_s) + \hat{b}_s(I_s) + (t - s)\nabla \hat{v}_{s,t}(I_s)\hat{b}_s(I_s)|^2\right] ds dt.$$

(37)

We consider a variant that avoids backpropagating through any spatial or temporal gradient

$$\mathcal{L}_{\mathsf{EMD}}(\hat{v}; \hat{b})$$

(38)

$$= \int_0^1 \int_0^t \mathbb{E}_{\rho_s}\left[|-\hat{v}_{s,t}(I_s) + \mathsf{sg}\left((t - s)\partial_s \hat{v}_{s,t}(I_s) + \hat{b}_s(I_s) + (t - s)\nabla \hat{v}_{s,t}(I_s)\hat{b}_s(I_s)\right)|^2\right] ds dt.$$

This yields the semigradient,

$$\frac{\delta \mathcal{L}_{\mathsf{EMD}}}{\delta \hat{v}_{s,t}}(\hat{v}; \hat{b})$$
$$= \hat{v}_{s,t}(I_s) - (t-s)\partial_s \hat{v}_{s,t}(I_s) - \hat{b}_s(I_s) - (t-s)\nabla \hat{v}_{s,t}(I_s)\hat{b}_s(I_s), \tag{39}$$
$$= -\left(\partial_s \hat{X}_{s,t}(I_s) + \nabla \hat{X}_{s,t}(I_s)\hat{b}_s(I_s)\right).$$

In the last line, we applied (36), which agrees with (35). Hence, the objective (38) is equivalent to the consistency distillation objective in the continuous-time limit after a suitable rescaling of gradients. We note that (39) is identical to the "Align Your Flow" update considered by Sabour et al. (2025).

### C.2 Consistency training and mean flow.

Consistency training aims to train a model directly, avoiding access to a pre-trained teacher (Song and Dhariwal, 2023; Lu and Song, 2024). The associated loss follows from an identical derivation, except it uses two points on the same interpolant trajectory (rather than $\hat{x}_{s+\Delta s}$, which requires access to $\hat{b}_s$),

$$I_s = \alpha_s x_0 + \beta_s x_1,$$
$$I_{s+\Delta s} = \alpha_{s+\Delta s} x_0 + \beta_{s+\Delta s} x_1 = I_s + \Delta s \dot{I}_s + o(\Delta s). \tag{40}$$

In (40), $x_0$ and $x_1$ are shared between $I_s$ and $I_{s+\Delta s}$, which yields the second equality for $I_{s+\Delta s}$. Following the same steps as for consistency distillation, the final result is to replace the semi-gradient (39) by a Monte-Carlo approximation that leverages $\dot{I}_s$ in place of the *true* vector field $b_s(x) = \mathbb{E}\left[\dot{I}_s \mid I_s = x\right]$,

$$\nabla_{\mathsf{CT}} = -\left(\partial_s \hat{X}_{s,t}(I_s) + \nabla \hat{X}_{s,t}(I_s)\dot{I}_s\right). \tag{41}$$

For the parameterization (6), (41) becomes

$$\nabla_{\mathsf{CT}} = \hat{v}_{s,t}(I_s) - (t-s)\partial_s \hat{v}_{s,t}(I_s) - \dot{I}_s - (t-s)\nabla \hat{v}_{s,t}(I_s)\dot{I}_s. \tag{42}$$

The semigradients (41) and (42) are higher-variance than (35) as Monte-Carlo approximations, but on average give access to the ideal flow $b$ rather than the pre-trained, approximate flow $\hat{b}$. The gradient (42) is identical to the "mean flow" update recently considered by Geng et al. (2025).

## D  Connection to shortcut models.

Shortcut models (Frans et al., 2024) correspond to a subset of our proposed PSD scheme (15), which itself is based on PFMM. To touch base with the formulation of PFMM in (31), as well as the discussion of PSD in the main text, we place shortcut models in our notation here.

Shortcut models consider a fixed grid of times $0 = t_0 < t_1 < \ldots < t_N = 1$ spaced dyadically, so that $t_{i+2} - t_{i+1} = 2(t_{i+1} - t_i)$. Observing a similar relation to the tangent identity (7), they train $\hat{v}_{t,t}$ like a flow matching model and leverage (31) as a bootstrapping mechanism,

$$\mathcal{L}_{\mathsf{S}}(\hat{X}) = \int_0^1 \mathbb{E}\left[|\hat{v}_{t,t} - \dot{I}_t|^2\right] + \mathbb{E}\left[|\hat{X}_{t_i,t_{i+2}}(I_{t_i}) - \mathsf{sg}\left(\hat{X}_{t_{i+1},t_{i+2}}\left(\hat{X}_{t_i,t_{i+1}}(I_{t_i})\right)\right)|^2\right], \tag{43}$$
$$\hat{X}_{s,t}(x) = x + (t-s)\hat{v}_{s,t}(x).$$

Clearly, the second term in (43) reduces to (31) with $s, t, u$ restricted to a fixed grid. Similarly, (43) corresponds to (15) with discretization in time and the specific proposal distribution $p_u^i = \delta_{t_i}$. The second term in (43) can also be written entirely in terms of $\hat{v}$ using the preconditioning discussed later in Section F.1, which leads to the exact form of the objective discussed in Frans et al. (2024).

## E  Proofs

In this work, we assume that all studied differential equations satisfy the following assumption.

**Assumption E.1.** *The drift satisfies the one-sided Lipschitz condition*

$$\exists\ C > 0 : \quad (b_t(x) - b_t(y)) \cdot (x - y) \leqslant C|x - y|^2 \quad \textit{for all } (t, x, y) \in [0, 1] \times \mathbb{R}^d \times \mathbb{R}^d. \quad (44)$$

Under Assumption E.1, the classical Cauchy-Lipschitz theory guarantees that solutions exist and are unique for all $x_0 \in \mathbb{R}^d$ and for all $t \in [0, 1]$.

We first provide a self-contained proof of the following proposition, which first appeared in Boffi et al. (2024). We will then apply this result to prove the primary claims of the main text.

**Proposition E.2.** *Let $X_{s,t}$ denote the flow map (4) for the probability flow equation $\dot{x}_t = b_t(x_t)$. Then $X_{s,t}$ satisfies the Lagrangian equation,*

$$\partial_t X_{s,t}(x) = b_t(X_{s,t}(x)), \qquad \forall\ (x, s, t) \in \mathbb{R}^d \times [0, 1]^2 \quad (45)$$

*the Eulerian equation,*

$$\partial_s X_{s,t}(x) + \nabla X_{s,t}(x) b_s(x) = 0, \qquad \forall\ (x, s, t) \in \mathbb{R}^d \times [0, 1]^2, \quad (46)$$

*and the semigroup property*

$$X_{s,t}(x) = X_{u,t}(X_{s,u}(x)) \qquad \forall\ (x, u, s, t) \in \mathbb{R}^d \times [0, 1]^3. \quad (47)$$

*Proof.* Repeating (4) for ease of reading, the flow map satisfies the jump condition

$$X_{s,t}(x_s) = x_t, \qquad \forall\ (s, t) \in [0, 1]^2, \quad (48)$$

where $x_t$ denotes a trajectory of the probability flow (2). The proof of each condition relies on careful manipulation of this equation.

We first prove the semigroup condition. Observe that

$$X_{u,t}(X_{s,u}(x_s)) = X_{u,t}(x_u) = x_t = X_{s,t}(x_s) \quad (49)$$

Because $x_s$ was arbitrary, the result follows.

We now prove the Lagrangian condition. Taking a derivative of (48), with respect to $t$ and applying the probability flow (2), we find

$$\begin{aligned} \partial_t X_{s,t}(x_s) &= \dot{x}_t, \\ &= b_t(x_t), \\ &= b_t(X_{s,t}(x_s)). \end{aligned} \quad (50)$$

Because $x_s$ was arbitrary, we obtain the Lagrangian condition (45)

Last, we prove the Eulerian condition. Taking a total derivative of (48) with respect to $s$, we find that

$$\begin{aligned} \frac{d}{ds} X_{s,t}(x_s) &= \partial_s X_{s,t}(x_s) + \nabla X_{s,t}(x_s) \dot{x}_s, \\ &= \partial_s X_{s,t}(x_s) + \nabla X_{s,t}(x_s) b_s(x_s) \end{aligned} \quad (51)$$

Again, because $x_s$ was arbitrary, the result follows. $\square$

We now provide a simple proof of the tangent condition we leverage in the main text.

**Lemma 2.1** (Tangent condition)**.** *Let $X_{s,t}$ denote the flow map. Then,*

$$\lim_{s \to t} \partial_t X_{s,t}(x) = b_t(x) \qquad \forall t \in [0, 1], \quad \forall x \in \mathbb{R}^d, \quad (5)$$

*i.e. the tangent vectors to the curve $(X_{s,t}(x))_{t \in [s,1]}$ give the velocity field $b_t(x)$ for every $x$.*

*Proof.* By Proposition E.2, we have that the flow map satisfies the Lagrangian equation (45). Taking the limit as $s \to t$, and assuming continuity of the flow map, we find

$$\lim_{s \to t} \partial_t X_{s,t}(x) = \lim_{s \to t} b_t(X_{s,t}(x)) = b_t(X_{t,t}(x)) = b_t(x). \quad (52)$$

Above, we used that $X_{t,t}(x) = x$ for all $x \in \mathbb{R}^d$ and for all $t \in [0, 1]$. $\square$

We now prove Proposition 2.2, which extends Proposition E.2 to the representation (6).

**Proposition 2.2** (Flow map). *Assume that $X_{s,t}$ is given by (6) with $v_{s,t}$ satisfying (7), and assume that $v_{s,t}$ is continuous in both time arguments. Then, $X_{s,t}$ is the flow map defined in (4) if and only if any of the following conditions also holds:*

(i) *(Lagrangian condition): $X_{s,t}$ solves the Lagrangian equation*

$$\partial_t X_{s,t}(x) = v_{t,t}(X_{s,t}(x)), \tag{8}$$

*for all $(s,t) \in [0,1]^2$ and for all $x \in \mathbb{R}^d$.*

(ii) *(Eulerian condition): $X_{s,t}$ solves the Eulerian equation*

$$\partial_s X_{s,t}(x) + \nabla X_{s,t}(x) v_{s,s}(x) = 0, \tag{9}$$

*for all $(s,t) \in [0,1]^2$ and for all $x \in \mathbb{R}^d$.*

(iii) *(Semigroup condition): For all $(s,t,u) \in [0,1]^3$ and for all $x \in \mathbb{R}^d$,*

$$X_{u,t}(X_{s,u}(x)) = X_{s,t}(x). \tag{10}$$

*Proof.* We start with the Lagrangian condition (8). By assumption of (7), $v_{t,t}(x) = b_t(x)$, so that (8) is equivalent to (45). It follows that the flow map must satisfy (8) by Proposition E.2, which proves the forward implication. To prove the reverse implication, observe that by Assumption E.1, solutions to (8) are unique, so that any solution must be the flow map.

The proof of the Eulerian condition is similar. For the forward implication, we observe that (9) is equivalent to (46), so that the flow map solves (9). Now, let $X$ solve (9) (along with (6) and (7)). We would like to prove that $X$ is the flow map. Let us observe that by assumption,

$$\frac{d}{ds} X_{s,t}(x_s) = 0, \tag{53}$$

where $x_s$ is any solution of the probability flow. Integrating both sides with respect to $s$ from $s$ to $t$, we find that

$$X_{s,t}(x_s) - X_{t,t}(x_t) = 0 \implies x_t = X_{s,t}(x_s). \tag{54}$$

This is precisely the definition of the flow map.

Last, we prove the final property. By Proposition E.2, we have that the flow map satisfies (10), which proves the forward implication. To prove the reverse implication, let $X$ be any map satisfying (6), (7) and (10). Define the notation $\partial_t X_{t,t}(y) = \lim_{s \to t} \partial_t X_{s,t}(y) = v_{t,t}(y) = b_t(y)$. Then, consider a Taylor expansion of the infinitesimal semigroup condition for $(x,s,t) \in \mathbb{R}^d \times [0,1]^2$ arbitrary,

$$\begin{aligned} X_{s,t+h}(x) &= X_{t,t+h}(X_{s,t}(x)), \\ &= X_{t,t}(X_{s,t}(x)) + h \partial_t X_{t,t}(X_{s,t}(x)) + o(h), \\ &= X_{s,t}(x) + h \partial_t X_{t,t}(X_{s,t}(x)) + o(h), \\ &= X_{s,t}(x) + h v_{t,t}(X_{s,t}(x)) + o(h), \\ &= X_{s,t}(x) + h b_t(X_{s,t}(x)) + o(h). \end{aligned} \tag{55}$$

Note that the above Taylor expansion implicitly uses that $v_{s,t}$ is continuous in $(s,t)$ to write $v_{t,t+h} = v_{t,t} + O(h)$. This rules out the discontinuous solution

$$v_{s,t}(x) = \begin{cases} b_t(x) & s = t, \\ 0 & s \neq t, \end{cases} \tag{56}$$

which corresponds to $X_{s,t}(x) = x$ for all $(x,s,t) \in \mathbb{R}^d \times [0,1]^2$ and satisfies the semigroup condition trivially.

Re-arranging the last line of (55), we find that

$$\frac{X_{s,t+h}(x) - X_{s,t}(x)}{h} = b_t(X_{s,t}(x)) + o(1), \tag{57}$$

so that

$$\lim_{h \to 0} \frac{X_{s,t+h}(x) - X_{s,t}(x)}{h} = \partial_t X_{s,t}(x) = b_t(X_{s,t}(x)). \tag{58}$$

Equation (58) is precisely the Lagrangian equation, whose unique solution is the ideal flow map. This completes the proof. $\qquad \square$

Given the above developments, we now recall our main proposition.

**Proposition 2.3** (Self-distillation). *The flow map $X_{s,t}$ defined in (4) is given for all $0 \leqslant s \leqslant t \leqslant 1$ by $X_{s,t}(x) = x + (t - s)v_{s,t}(x)$ where $v_{s,t}(x)$ the unique minimizer over $\hat{v}$ of*

$$\mathcal{L}_{\mathsf{SD}}(\hat{v}) = \mathcal{L}_b(\hat{v}) + \mathcal{L}_{\mathsf{D}}(\hat{v}), \tag{11}$$

*where $\mathcal{L}_b(\hat{v})$ is given by*

$$\mathcal{L}_b(\hat{v}) = \int_0^1 \mathbb{E}_{x_0,x_1}\big[|\hat{v}_{t,t}(I_t) - \dot{I}_t|^2\big]dt, \tag{12}$$

*and where $\mathcal{L}_{\mathsf{D}}(\hat{v})$ is any of the following three objectives.*

*(i) The* Lagrangian self-distillation (LSD) *objective, which leverages (8),*

$$\mathcal{L}_{\mathsf{LSD}}(\hat{v}) = \int_0^1 \int_0^t \mathbb{E}_{x_0,x_1}\big[\big|\partial_t \hat{X}_{s,t}(I_s) - \hat{v}_{t,t}(\hat{X}_{s,t}(I_s))\big|^2\big]dsdt; \tag{13}$$

*(ii) The* Eulerian self-distillation (ESD) *objective, which leverages (9),*

$$\mathcal{L}_{\mathsf{ESD}}(\hat{v}) = \int_0^1 \int_0^t \mathbb{E}_{x_0,x_1}\big[\big|\partial_s \hat{X}_{s,t}(I_s) + \nabla \hat{X}_{s,t}(I_s)\hat{v}_{s,s}(I_s)\big|^2\big]dsdt; \tag{14}$$

*(iii) The* progressive self-distillation (PSD) *objective, which leverages (10),*

$$\mathcal{L}_{\mathsf{PSD}}(\hat{v}) = \int_0^1 \int_0^t \int_s^t \mathbb{E}_{x_0,x_1}\big[\big|\hat{X}_{s,t}(I_s) - \hat{X}_{u,t}(\hat{X}_{s,u}(I_s))\big|^2\big]dudsdt. \tag{15}$$

*Above, $\hat{X}_{s,t}(x) = x + (t - s)\hat{v}_{s,t}(x)$ and $\mathbb{E}_{x_0,x_1}$ denotes an expectation over the random draws of $(x_0, x_1)$ in the interpolant defined in (1).*

*Proof.* We first prove the statement for the LSD algorithm. Observe that for any $\hat{b}_t$ and any $\hat{X}_{s,t}(x) = x + (t - s)\hat{v}_{s,t}(x)$,

$$\begin{aligned} \mathcal{L}_b(\hat{b}) &\geqslant \mathcal{L}_b(b), \\ \mathcal{L}_{\mathsf{LSD}}(\hat{v}) &\geqslant 0. \end{aligned} \tag{59}$$

where $b_t(x) = \mathbb{E}[\dot{I}_t | I_t = x]$ is the ideal flow. This follows because $\mathcal{L}_b$ is convex in $\hat{b}$ with unique global minimizer given by $b$, while $\mathcal{L}_{\mathsf{LSD}}$ is a square residual term on the Lagrangian relation (8). From this, we conclude

$$\mathcal{L}_{\mathsf{SD}}(\hat{v}) = \mathcal{L}_b(\hat{v}) + \mathcal{L}_{\mathsf{LSD}}(\hat{v}) \geqslant \mathcal{L}_b(b). \tag{60}$$

By Lemma 2.1 and Proposition 2.2, the ideal flow map $X_{s,t}$ satisfies

$$\begin{aligned} v_{t,t}(x) &= b_t(x) & \forall\, (x, t) \in \mathbb{R}^d \times [0, 1], \\ \partial_t X_{s,t}(x) &= v_{t,t}(X_{s,t}(x)) & \forall\, (x, s, t) \in \mathbb{R}^d \times [0, 1]^2. \end{aligned} \tag{61}$$

From (61), we see that

$$\mathcal{L}_{\mathsf{SD}}(X) = \mathcal{L}_b(v) + \mathcal{L}_{\mathsf{LSD}} = \mathcal{L}_b(v) = \mathcal{L}_b(b), \tag{62}$$

so that $X_{s,t}$ achieves the lower bound (60) and is therefore optimal. Moreover, any global minimizer must satisfy (61), and by Proposition 2.2 therefore must be the flow map.

We now prove the statement for the ESD algorithm, which is similar. We first observe that for any $\hat{v}$,

$$\begin{aligned} \mathcal{L}_b(\hat{v}) &\geqslant \mathcal{L}_b(b), \\ \mathcal{L}_{\mathsf{ESD}}(\hat{v}) &\geqslant 0. \end{aligned} \tag{63}$$

From above, we conclude

$$\mathcal{L}_{\mathsf{SD}}(\hat{v}) = \mathcal{L}_b(\hat{v}) + \mathcal{L}_{\mathsf{ESD}}(\hat{v}) \geqslant \mathcal{L}_b(b). \tag{64}$$

Moreover, by Lemma 2.1 and Proposition 2.2,

$$\begin{aligned} v_{t,t}(x) &= b_t(x) & \forall\, (x, t) \in \mathbb{R}^d \times [0, 1], \\ \partial_s X_{s,t}(x) &= -\nabla X_{s,t}(x)v_{s,s} & \forall\, (x, s, t) \in \mathbb{R}^d \times [0, 1]^2. \end{aligned} \tag{65}$$

From (65), it follows that the ideal flow map satisfies

$$\mathcal{L}_{\mathsf{SD}}(v) = \mathcal{L}_b(v) + \mathcal{L}_{\mathsf{ESD}}(v) = \mathcal{L}_b(v) = \mathcal{L}_b(b). \tag{66}$$

Equation (66) shows that $X_{s,t}$ achieves the lower bound (64) and hence is optimal. Moreover, any global minimizer must satisfy (65) and therefore by Proposition 2.2 is the ideal flow map.

Finally, we prove the result for the PSD approach. The proposition is stated for a uniform proposal distribution over $u$, but holds for any distribution with full support over $[s, t]$. First, we observe that

$$\mathcal{L}_{\mathsf{SD}}(\hat{v}) = \mathcal{L}_b(\hat{v}) + \mathcal{L}_{\mathsf{PSD}}(\hat{v}) \geqslant \mathcal{L}_b(b). \tag{67}$$

By the semigroup property (10), we have that the true flow map satisfies

$$\mathcal{L}_{\mathsf{SD}}(v) = \mathcal{L}_b(v) + \mathcal{L}_{\mathsf{PSD}}(v) = \mathcal{L}_b(v) = \mathcal{L}_b(b), \tag{68}$$

so that $X$ is optimal. Now, let $X^*$ be any map satisfying

$$\mathcal{L}_{\mathsf{SD}}(X^*) = \mathcal{L}_b(b), \tag{69}$$

i.e., any global minimizer of the PSD objective. It then necessarily follows that

$$\begin{aligned} \mathcal{L}_b(v^*) &= \mathcal{L}_b(v), \\ \mathcal{L}_{\mathsf{PSD}}(v^*) &= 0. \end{aligned} \tag{70}$$

By Proposition 2.2, under the assumption that $v^*$ is continuous, (70) implies that $X^*$ is the ideal flow map $X$. $\qquad\square$

We now recall our theoretical error bounds for the LSD and ESD algorithms.

**Proposition 2.4** (Wasserstein bounds). *Let $\hat{X}_{s,t}(x) = x + (t - s)\hat{v}_{s,t}(x)$ denote a candidate flow map, let $\hat{\rho}_1 = \hat{X}_{0,1}\sharp\rho_0$ denote the corresponding one-step generated distribution, and let $\hat{L}$ denote the spatial Lipschitz constant of $\hat{v}_{t,t}(\cdot)$ uniformly in $t$. First assume $\mathcal{L}_b(\hat{v}) + \mathcal{L}_{\mathsf{LSD}}(\hat{v}) \leqslant \varepsilon$. Then,*

$$\mathsf{W}_2^2(\hat{\rho}_1, \rho_1) \leqslant 4e^{1+2\hat{L}}\varepsilon. \tag{16}$$

*Now assume that $\mathcal{L}_b(\hat{v}) + \mathcal{L}_{\mathsf{ESD}}(\hat{v}) \leqslant \varepsilon$. Then,*

$$\mathsf{W}_2^2(\hat{\rho}_1, \rho_1) \leqslant 2e \cdot (1 + e^{2\hat{L}})\varepsilon. \tag{17}$$

For ease of reading, we split the proof of Proposition 2.4 into two results, one for each algorithm. We begin with LSD.

**Proposition E.3** (Lagrangian self-distillation). *Consider the Lagrangian self-distillation method,*

$$\mathcal{L}_{\mathsf{SD}}(\hat{X}) = \mathcal{L}_b(\hat{v}) + \mathcal{L}_{\mathsf{LSD}}(\hat{v}). \tag{71}$$

*Let $\hat{X}$ denote a candidate flow map satisfying $\mathcal{L}_{\mathsf{SD}}(\hat{X}) \leqslant \varepsilon$, and let $\hat{\rho}_1$ denote the corresponding pushforward $\hat{\rho}_1 = \hat{X}_{0,1}\sharp\rho_0$. Let $\hat{L}$ denote the spatial Lipschitz constant of $\hat{v}_{t,t}(\cdot)$ uniformly in time, i.e.*

$$|\hat{v}_{t,t}(x) - \hat{v}_{t,t}(y)| \leqslant \hat{L}|x - y| \qquad \forall\ (x, y, t) \in \mathbb{R}^d \times \mathbb{R}^d \times [0, 1]. \tag{72}$$

*Then,*

$$\mathsf{W}_2^2(\hat{\rho}_1, \rho_1) \leqslant 4e^{1+2\hat{L}}\varepsilon. \tag{73}$$

*Proof.* Observe that $\mathcal{L}_{\mathsf{SD}}(\hat{X}) \leqslant \varepsilon$ implies that both $\mathcal{L}_b(\hat{v}) \leqslant \varepsilon$ and $\mathcal{L}_{\mathsf{LSD}}(\hat{v}) \leqslant \varepsilon$. We first note that

$$\begin{aligned} \mathcal{L}_b(\hat{v}) &= \int_0^1 \mathbb{E}_{\rho_t}\left[|\hat{v}_{t,t}(I_t) - \dot{I}_t|^2\right] dt, \\ &= \int_0^1 \mathbb{E}_{\rho_t}\left[|\hat{v}_{t,t}(I_t) - b_t(I_t)|^2\right] dt + \int_0^1 \mathbb{E}_{\rho_t}\left[|\dot{I}_t|^2 - |b_t(I_t)|^2\right] dt. \end{aligned} \tag{74}$$

In (74), we used that $b_t(x) = \mathbb{E}[\dot{I}_t | I_t = x]$ along with the tower property of the conditional expectation. It then follows that the $L_2$ error from the target flow $b$ is bounded by

$$\int_0^1 \mathbb{E}_{\rho_t}\left[|\hat{v}_{t,t}(I_t) - b_t(I_t)|^2\right] dt \leqslant \varepsilon - \int_0^1 \mathbb{E}_{\rho_t}\left[|\dot{I}_t|^2 - |b_t(I_t)|^2\right] dt. \tag{75}$$

We now observe that, again by the tower property of the conditional expectation,

$$
\begin{aligned}
\int_0^1 \mathbb{E}_{\rho_t}\left[|\dot{I}_t|^2 - |b_t(I_t)|^2\right] dt &= \int_0^1 \mathbb{E}_{\rho_t}\left[\mathbb{E}\left[|\dot{I}_t|^2 - |b_t(I_t)|^2 \mid I_t\right]\right] dt, \\
&= \int_0^1 \mathbb{E}_{\rho_t}\left[\mathbb{E}\left[|\dot{I}_t - b_t(I_t)|^2 \mid I_t\right]\right] dt, \\
&\geqslant 0.
\end{aligned}
\tag{76}
$$

Equation (76) shows that the term subtracted in (75) is a conditional variance, and therefore is nonnegative. Combining the two, we find that

$$
\int_0^1 \mathbb{E}_{\rho_t}\left[|\hat{v}_{t,t}(I_t) - b_t(I_t)|^2\right] \leqslant \varepsilon.
\tag{77}
$$

We now consider the learned probability flow

$$
\dot{\hat{x}}_t^{\hat{v}} = \hat{v}_{t,t}\left(\hat{x}_t^{\hat{v}}\right), \qquad \hat{x}_0 \sim \rho_0.
\tag{78}
$$

By Proposition 3 of Albergo and Vanden-Eijnden (2022), (77) implies that

$$
\mathsf{W}_2^2\left(\rho_1, \hat{\rho}_1^v\right) \leqslant e^{1+2\hat{L}}\varepsilon.
\tag{79}
$$

where $\hat{\rho}_1^{\hat{v}} = \mathsf{Law}(\hat{x}_t^{\hat{v}})$. Now, by Proposition 3.7 of Boffi et al. (2024), $\mathcal{L}_{\mathsf{LSD}}(\hat{v}) \leqslant \varepsilon$ implies

$$
\mathsf{W}_2^2\left(\hat{\rho}_1^{\hat{v}}, \hat{\rho}_1\right) \leqslant e^{1+2\hat{L}}\varepsilon.
\tag{80}
$$

By the triangle inequality and Young's inequality, we then have

$$
\begin{aligned}
\mathsf{W}_2^2\left(\rho_1, \hat{\rho}_1\right) &\leqslant 2\left(\mathsf{W}_2^2\left(\rho_1, \hat{\rho}_1^{\hat{v}}\right) + \mathsf{W}_2^2\left(\hat{\rho}_1^{\hat{v}}, \hat{\rho}_1\right)\right), \\
&\leqslant 4e^{1+2\hat{L}}\varepsilon.
\end{aligned}
\tag{81}
$$

This completes the proof. $\qquad\square$

We now prove a similar guarantee for the ESD method.

**Proposition E.4** (Eulerian self-distillation). *Consider the Eulerian self-distillation method,*

$$
\mathcal{L}_{\mathsf{SD}}(\hat{v}) = \mathcal{L}_b(\hat{v}) + \mathcal{L}_{\mathsf{ESD}}(\hat{v}).
\tag{82}
$$

*Let $\hat{X}$ denote a candidate flow map with the same properties as in Proposition E.3. Then,*

$$
\mathsf{W}_2^2\left(\hat{\rho}_1, \rho_1\right) \leqslant 2e(1 + e^{2\hat{L}})\varepsilon.
\tag{83}
$$

*Proof.* As in Proposition E.3, our assumption $\mathcal{L}_{\mathsf{SD}}(\hat{v}) \leqslant \varepsilon$ implies that both $\mathcal{L}_b(\hat{v}) \leqslant \varepsilon$ and $\mathcal{L}_{\mathsf{ESD}}(\hat{v}) \leqslant \varepsilon$. Defining the flow $\dot{\hat{x}}_t^{\hat{v}}$ as in (78), we have a bound identical to (79) on $\mathsf{W}_2^2\left(\rho_1, \hat{\rho}_1^{\hat{v}}\right)$. Now, leveraging Proposition 3.8 in Boffi et al. (2024), we have that

$$
\mathsf{W}_2^2\left(\hat{\rho}_1^{\hat{v}}, \hat{\rho}_1\right) \leqslant e\varepsilon.
\tag{84}
$$

Again applying the triangle inequality and Young's inequality yields the relation

$$
\begin{aligned}
\mathsf{W}_2^2\left(\rho_1, \hat{\rho}_1\right) &\leqslant 2\left(\mathsf{W}_2^2\left(\hat{\rho}_1^{\hat{v}}, \hat{\rho}_1\right) + \mathsf{W}_2^2\left(\rho_1, \hat{\rho}_1^{\hat{v}}\right)\right), \\
&\leqslant 2e(1 + e^{2\hat{L}})\varepsilon.
\end{aligned}
\tag{85}
$$

This completes the proof. $\qquad\square$

# F  Further details on self-distillation

In this section, we collect some additional results and detail on some of the topics discussed in the main text.

## F.1 Semigroup parameterization for PSD.

By definition, we have that

$$X_{s,t}(x) = x + (t - s)v_{s,t}(x).\tag{86}$$

We then also have that

$$X_{s,u}(x) = x + (u - s)v_{s,u}(x),$$
$$X_{u,t}(X_{s,u}(x)) = X_{s,u}(x) + (t - u)v_{u,t}(X_{s,u}(x)),\tag{87}$$
$$= x + (u - s)v_{s,u}(x) + (t - u)v_{u,t}(X_{s,u}(x)).$$

By the semigroup property (10), it follows that

$$X_{s,t}(x) = X_{u,t}(X_{s,u}(x))\tag{88}$$

from which we see that

$$x + (t - s)v_{s,t}(x) = x + (u - s)v_{s,u}(x) + (t - u)v_{u,t}(X_{s,u}(x)).\tag{89}$$

Re-arranging and eliminating, we find that

$$v_{s,t}(x) = \left(\frac{u - s}{t - s}\right)v_{s,u}(x) + \left(\frac{t - u}{t - s}\right)v_{u,t}(X_{s,u}(x)),\tag{90}$$

which provides a direct signal for $v_{s,t}$. Choosing $u = \gamma s + (1 - \gamma)t$ for $\gamma \in [0, 1]$ leads to the simple relations

$$\frac{u - s}{t - s} = 1 - \gamma, \qquad \frac{t - u}{t - s} = \gamma,\tag{91}$$

which can be used to precondition the relation (90) as

$$v_{s,t}(x) = (1 - \gamma)v_{s,u}(x) + \gamma v_{u,t}(X_{s,u}(x)).\tag{92}$$

In the numerical experiments, we use (92) to define a training signal for $\hat{v}$ for the PSD algorithm.

## F.2 Limiting relations and annealing schemes.

**Limiting relations.** As shown in the proof of Proposition 2.2, application of the semigroup property with $(s, u, t) = (s, t, t + h)$ for a fixed $(s, t)$ recovers the Lagrangian equation at order $h$. As shown in the proof of the tangent condition Lemma 2.1, the Lagrangian condition recovers the velocity field in the limit as $s \to t$. Similarly, if we consider the Eulerian equation in the limit as $s \to t$,

$$\lim_{t \to s} \partial_s X_{s,t}(x) + \nabla X_{s,t}(x)b_s(x) = \partial_s X_{s,s}(x) + b_s(x) = 0,\tag{93}$$

so that $\partial_s X_{s,s}(x) = -v_{s,s}(x) = -b_s(x)$. In this way, all three characterizations reduce to the flow matching objective for $v_{t,t}$ as the diagonal is approached.

**Annealing and pre-training.** As a result of (93), we can view training the flow $\hat{v}_{t,t}$ only on the diagonal $s = t$ as a pre-training scheme for the map $\hat{X}$. This also means that we can initialize $\hat{v}_{t,t}$ from a pre-trained model in a principled way via appropriate duplication of the time embeddings.

The relations (7) and (93) imply that the off-diagonal self-distillation terms represent a natural extension of the diagonal flow matching term. This suggests a simple two-phase curriculum in which the flow matching term is trained alone for $N_{\mathrm{fm}}$ steps as a pre-training phase, followed by a smooth conversion from diagonal training into self-distillation by expanding the sampled range of $|t - s|$ from 0 to 1 over the course of $N_{\mathrm{anneal}}$ steps. This can be accomplished, for example, by drawing $(s, t)$ uniformly on the off-diagonal and then clamping $t = \min(t, s + \delta(k))$ where $k$ denotes the iteration and $\delta(k)$ is the maximum value of $|t - s|$, for example $\delta(k) = k/N_{\mathrm{anneal}}$. For simplicity, we trained directly without any annealing in our experiments, but expect this to simplify and speed up training for large datasets where overfitting is not a concern.

## F.3 Further details on loss sampling and computation

In this section, we provide further detail on how the choice of $\eta \in [0, 1]$, which distributes the batch between the diagonal flow matching term and the off-diagonal self-distillation term, affects training time. The factor $\eta$ can be chosen based on the available computational budget to systematically trade off the relative amount of direct training and distillation per gradient step. We focus here on the computational cost of a forward pass of the objective function; the complexity of a backward pass will depend on the specific choice of $\mathrm{sg}\,(\cdot)$ operator used.

**Flow matching.** Evaluating the interpolant loss $\mathcal{L}_b$ on a single sample requires a single neural network evaluation $\hat{v}_{t,t}(I_t)$, leading to $B$ network evaluations on a batch.

**LSD.** The LSD objective requires a single partial derivative evaluation $\partial_t \hat{X}_{s,t}(I_s)$ and two network evaluations – one for $\hat{v}_{t,t}$ and one for $\hat{X}_{s,t}(I_s)$ – per sample. The time derivative is a constant factor $C \approx 1.5$ more than a forward pass, and with standard computational tools such as jvp, can be computed at the same time as $\hat{X}_{s,t}(I_s)$. The LSD objective thus requires $(1 + C)B$ network evaluations. Adding the diagonal and off-diagonal parts, we find a complexity of $((1-\eta)(1+C)+\eta)B$ for the full self-distillation objective.

**PSD.** The PSD objective requires three neural network evaluations, so that its expense is $3B$. Combining this with the diagonal component, we have $(3(1 - \eta) + \eta)B$ network evaluations.

**ESD.** The ESD objective requires a partial derivative evaluation $\partial_s X_{s,t}(I_s)$, a neural network evaluation $v_{s,s}(I_s)$, and a Jacobian-vector product $\nabla X_{s,t}(I_s)v_{s,s}(I_s)$. Observing that $(\partial_s, \nabla)$ can be used as one augmented $(d + 1)$-dimensional gradient, and then observing that $\partial_s \hat{X}_{s,t}(I_s) + \nabla \hat{X}_{s,t}(I_s)\hat{v}_{s,s}(I_s) = \nabla_{s,x}\hat{X}_{s,t}(I_s)\begin{pmatrix} 1 \\ \hat{v}_{s,s}(I_s) \end{pmatrix}$, this can be computed as a single Jacobian-vector product. This gives a complexity of $(1 + C)B$, identical to LSD. Adding the diagonal component, we find $((1 + C)(1 - \eta) + \eta)B$.

### F.4 Stopgradient recommendations

The choice of $\mathsf{sg}\,(\cdot)$ operator in the loss is delicate and empirical, as it is very difficult to ascertain the convergence properties of an algorithm operating on an objective leveraging $\mathsf{sg}\,(\cdot)$ *a-priori*. Nevertheless, in practice, we find it critical for high-dimensional tasks such as images to use $\mathsf{sg}\,(\cdot)$ to control the flow of information from the teacher network on the diagonal $s = t$ to the off-diagonal. For large-scale neural networks, we find empirically that backpropagating through Jacobian-vector products – in particular spatial Jacobian-vector products – leads to significant instability, which can be avoided with $\mathsf{sg}\,(\cdot)$. For low-dimensional tasks with simple neural networks, we found instability to be less of a concern.

Following these observations, we found it useful to take insight from the distillation setting described in Section B, leading to the configurations

$$\mathcal{L}_{\mathsf{LSD}}(\hat{v}) = \int_0^1 \int_0^t \mathbb{E}\left[|\partial_t \hat{X}_{s,t}(I_s) - \mathsf{sg}\left(\hat{v}_{t,t}(\hat{X}_{s,t}(I_s))\right)|^2\right],$$

$$\mathcal{L}_{\mathsf{ESD}}(\hat{v}) = \int_0^1 \int_0^t \mathbb{E}\left[|\partial_s \hat{X}_{s,t}(I_s) + \mathsf{sg}\left(\nabla \hat{X}_{s,t}(I_s)\hat{v}_{s,s}(I_s)\right)|^2\right], \tag{94}$$

$$\mathcal{L}_{\mathsf{PSD}}(\hat{v}) = \int_0^1 \int_0^t \mathbb{E}_{p_\gamma}\mathbb{E}_{I_s}\left[|\hat{v}_{s,t}(I_s) - \mathsf{sg}\left((1 - \gamma)\hat{v}_{s,u}(I_s) + \gamma\hat{v}_{u,t}(\hat{X}_{s,u}(I_s))\right)|^2\right].$$

It is also possible to avoid backpropagating through the partial derivative with respect to $s$ and with respect to $t$ in ESD and LSD by expanding the definition of $\hat{X}_{s,t}(x) = x + (t-s)\hat{v}_{s,t}(x)$ as described in Section C. This reduces the memory overhead even further by avoiding backpropagating through a backward pass of the network.

**EMA teacher.** In addition to the use of $\mathsf{sg}\,(\cdot)$, an important consideration is the choice of parameters for the teacher, which provides an alternative perspective on and method to implement the $\mathsf{sg}\,(\cdot)$. Making explicit the student parameters $\theta$ and teacher parameters $\phi$, we can write for $\mathcal{L}_{\mathsf{LSD}}$ (with analogous expressions for the other choices),

$$\mathcal{L}_{\mathsf{LSD}}(\hat{v}) = \int_0^1 \int_0^t \mathbb{E}\left[|\partial_t \hat{X}_{s,t}^\theta(I_s) - \hat{v}_{t,t}^\phi(\hat{X}_{s,t}^\phi(I_s))|^2\right]. \tag{95}$$

The recommendation in (94) corresponds to taking the gradient of (95) with respect to $\theta$ and then evaluating the result at $\phi = \theta$. A second option would be to evaluate $\phi$ at an exponential moving average of $\theta$,

$$\phi_k = \delta\phi_{k-1} + (1 - \delta)\theta_k, \qquad \delta \in [0, 1], \tag{96}$$

where $k$ denotes the optimization step and where $\delta$ denotes a forgetting factor such as $\delta = 0.9999$. While in practice we found improved samples by evaluating the learned flow map over EMA parameters (see Section G for exact values), we found that the use of EMA for the teacher parameters offered no gain and sometimes led to instability in early experiments. For this reason, we use the instantaneous parameters $\phi = \theta$, corresponding to (94) or $\delta = 0$ in (96).

## F.5 Classifier-free guidance

In this section, we describe how to train a flow map with classifier-free guidance. For the derivation, we focus on the LSD algorithm to avoid replicating the loss functions in each case, but the other choices are identical. To this end, let $b_t(x; c)$ denote a conditional velocity field. We first observe that we may train a conditional flow map via the objective function

$$\mathcal{L}^c(\hat{v}) = \int_0^1 \mathbb{E}\left[|\hat{v}_{t,t}(x; c) - \dot{I}_t^c|^2\right] + \int_0^1 \int_0^t \mathbb{E}\left[|\partial_t \hat{X}_{s,t}(I_s^c; c) - \mathsf{sg}\left(\hat{v}_{t,t}(\hat{X}_{s,t}(I_s^c; c); c)\right)|^2\right] \quad (97)$$

In (97), $I_t^c$ denotes the conditional interpolant (i.e., with $I_t^c = \alpha(t)x_0 + \beta(t)x_1^c$ with $x_1^c \sim \rho_1(\cdot \mid c)$ drawn conditionally on $c$), and $\mathbb{E}$ now includes an expectation over the value of $c$. To train a model that is both conditional and unconditional, we may include $c = \varnothing$ in the expectation.

As in the main text, let us now define the CFG velocity field at guidance strength $\alpha \in \mathbb{R}$ as

$$\begin{aligned} q_t(x; \alpha, c) &= b_t(x; \varnothing) + \alpha\left(b_t(x; c) - b_t(x; \varnothing)\right), \\ &= v_{t,t}(x; \varnothing) + \alpha\left(v_{t,t}(x; c) - v_{t,t}(x; \varnothing)\right). \end{aligned} \quad (98)$$

Given the second line of (98), we define the current estimate of the guided velocity,

$$\hat{q}_t(x; \alpha, c) = \hat{v}_{t,t}(x; \varnothing) + \alpha\left(\hat{v}_{t,t}(x; c) - \hat{v}_{t,t}(x; \varnothing)\right). \quad (99)$$

We now observe that because (99) is constructed entirely in terms of the known $\hat{v}_{t,t}$, we only need to modify the self-distillation term rather than the flow matching term to train a CFG flow map. To this end, we self-distill the guided velocity $\hat{q}_t$ over a range of $\alpha$. This leads to the objective function

$$\begin{aligned} \mathcal{L}_{\mathsf{LSD}}^{\mathsf{CFG}}(\hat{v}) &= \int_0^1 \mathbb{E}\left[|\hat{v}_{t,t}(x; c) - \dot{I}_t^c|^2\right] \\ &\quad + \int_0^{\bar{\alpha}} \int_0^1 \int_0^t \mathbb{E}\left[|\partial_t \hat{X}_{s,t}(I_s^c; \alpha, c) - \mathsf{sg}\left(\hat{q}_{t,t}\left(\hat{X}_{s,t}\left(I_s^c; \alpha, c\right)\right)\right)|^2\right], \end{aligned} \quad (100)$$

where $\bar{\alpha}$ denotes a maximum guidance scale of interest. Following the same derivation, we may obtain the CFG ESD objective

$$\begin{aligned} \mathcal{L}_{\mathsf{ESD}}^{\mathsf{CFG}}(\hat{v}) &= \int_0^1 \mathbb{E}\left[|\hat{v}_{t,t}(x; c) - \dot{I}_t^c|^2\right] \\ &\quad + \int_0^{\bar{\alpha}} \int_0^1 \int_0^t \mathbb{E}\left[|\partial_s \hat{X}_{s,t}(I_s^c; \alpha, c) + \mathsf{sg}\left(\nabla \hat{X}_{s,t}(I_s^c; \alpha, c)\hat{q}_{s,s}\left(I_s^c; \alpha, c\right)\right)|^2\right], \end{aligned} \quad (101)$$

as well as the CFG PSD objective,

$$\begin{aligned} \mathcal{L}_{\mathsf{PSD}}^{\mathsf{CFG}}(\hat{v}) &= \int_0^1 \mathbb{E}\left[|\hat{v}_{t,t}(x; c) - \dot{I}_t^c|^2\right] \\ &\quad + \int_0^{\bar{\alpha}} \int_0^1 \int_0^t \mathbb{E}\left[|\hat{v}_{s,t}(I_s^c; \alpha, c) - \mathsf{sg}\left((1-\gamma)\hat{v}_{s,u}(I_s^c; \alpha, c) + \gamma\hat{v}_{u,t}(\hat{X}_{s,u}(I_s^c; \alpha, c); \alpha, c)\right)|^2\right]. \end{aligned} \quad (102)$$

## F.6 Detailed algorithms for each self-distillation method

Here, we provide detailed algorithmic implementations for each self-distillation method using the recommendations provided in (94). Each algorithm computes the flow matching loss $\mathcal{L}_b(\hat{v})$ over a batch of size $\eta M$ and the distillation loss $\mathcal{L}_{\mathsf{D}}(\hat{v})$ over a batch of size $(1 - \eta)M$, comprising a total batch size of $M$.

**Algorithm 2:** Lagrangian Self-Distillation (LSD)

---

**input:** Distribution $\rho(x_0, x_1)$; model $\hat{v}_{s,t}$; coefficients $\alpha_t, \beta_t$; batch size $M$; diagonal fraction $\eta$; weight $w_{s,t}$.

**repeat**

  Sample $M_d = \lfloor \eta M \rfloor$ pairs $(x_0^i, x_1^i) \sim \rho(x_0, x_1)$;
  Sample $M_d$ times $t_i \sim U([0, 1])$;
  Compute interpolants $I_{t_i} = \alpha_{t_i} x_0^i + \beta_{t_i} x_1^i$ and velocities, $\dot{I}_{t_i} = \dot{\alpha}_{t_i} x_0^i + \dot{\beta}_{t_i} x_1^i$;
  Compute diagonal loss $\mathcal{L}_b = \frac{1}{M_d} \sum_{i=1}^{M_d} e^{-w_{t_i,t_i}} |\hat{v}_{t_i,t_i}(I_{t_i}) - \dot{I}_{t_i}|^2 + w_{t_i,t_i}$;
  Sample $M_o = M - M_d$ pairs $(x_0^j, x_1^j) \sim \rho(x_0, x_1)$;
  Sample $M_o$ pairs $(s_j, t_j) \sim U_{\mathsf{od}}$;
  Compute interpolants: $I_{s_j} = \alpha_{s_j} x_0^j + \beta_{s_j} x_1^j$;
  Compute simultaneously via $\mathtt{jvp}$: $\hat{X}_{s_j,t_j}(I_{s_j}), \partial_t \hat{X}_{s_j,t_j}(I_{s_j})$;
  Evaluate teacher at transported point: $\hat{b}_{t_j} = \hat{v}_{t_j,t_j}(\hat{X}_{s_j,t_j}(I_{s_j}))$;
  Compute residual: $r_j = \partial_t \hat{X}_{s_j,t_j}(I_{s_j}) - \mathsf{sg}\left(\hat{b}_{t_j}\right)$;
  Compute LSD loss $\mathcal{L}_{\mathsf{LSD}} = \frac{1}{M_o} \sum_{j=1}^{M_o} \left(e^{-w_{s_j,t_j}} |r_j|^2 + w_{s_j,t_j}\right)$;
  Compute self-distillation loss $\mathcal{L}_{\mathsf{SD}} = \mathcal{L}_b + \mathcal{L}_{\mathsf{LSD}}$;
  Update $\hat{v}$ and $w$ using $\nabla \mathcal{L}_{\mathsf{SD}}$;

**until** *converged*;

**output:** Trained flow map $\hat{X}_{s,t}(x) = x + (t - s)\hat{v}_{s,t}(x)$

---

**Algorithm 3:** Eulerian Self-Distillation (ESD)

---

**input:** Distribution $\rho(x_0, x_1)$; model $\hat{v}_{s,t}$; coefficients $\alpha_t, \beta_t$; batch size $M$; diagonal fraction $\eta$; weight $w_{s,t}$.

**repeat**

  Sample $M_d = \lfloor \eta M \rfloor$ pairs $(x_0^i, x_1^i) \sim \rho(x_0, x_1)$;
  Sample $M_d$ times $t_i \sim U([0, 1])$;
  Compute interpolants $I_{t_i} = \alpha_{t_i} x_0^i + \beta_{t_i} x_1^i$ and velocities $\dot{I}_{t_i} = \dot{\alpha}_{t_i} x_0^i + \dot{\beta}_{t_i} x_1^i$;
  Compute diagonal loss: $\mathcal{L}_b = \frac{1}{M_d} \sum_{i=1}^{M_d} \left(e^{-w_{t_i,t_i}} |\hat{v}_{t_i,t_i}(I_{t_i}) - \dot{I}_{t_i}|^2 + w_{t_i,t_i}\right)$;
  Sample $M_o = M - M_d$ pairs $(x_0^j, x_1^j) \sim \rho(x_0, x_1)$;
  Sample $M_o$ pairs $(s_j, t_j) \sim U_{\mathsf{od}}$;
  Compute interpolants: $I_{s_j} = \alpha_{s_j} x_0^j + \beta_{s_j} x_1^j$;
  Evaluate teacher velocities: $\hat{b}_{s_j} = \hat{v}_{s_j,s_j}(I_{s_j})$;
  Compute simultaneously via single augmented $\mathtt{jvp}$: $\partial_s \hat{X}_{s_j,t_j}(I_{s_j}), \nabla \hat{X}_{s_j,t_j}(I_{s_j})$;
  Compute Eulerian residual: $r_j = \partial_s \hat{X}_{s_j,t_j}(I_{s_j}) + \mathsf{sg}\left(\nabla \hat{X}_{s_j,t_j}(I_{s_j}) \hat{b}_{s_j}\right)$;
  Compute ESD loss: $\mathcal{L}_{\mathsf{ESD}} = \frac{1}{M_o} \sum_{j=1}^{M_o} \left(e^{-w_{s_j,t_j}} |r_j|^2 + w_{s_j,t_j}\right)$;
  Compute self-distillation loss $\mathcal{L}_{\mathsf{SD}} = \mathcal{L}_b + \mathcal{L}_{\mathsf{ESD}}$;
  Update $\hat{v}$ and $w$ using $\nabla \mathcal{L}_{\mathsf{SD}}$;

**until** *converged*;

**output:** Trained flow map $\hat{X}_{s,t}(x) = x + (t - s)\hat{v}_{s,t}(x)$

**Algorithm 4:** Progressive Self-Distillation (PSD)

---

**input:** Distribution $\rho(x_0, x_1)$; model $\hat{v}_{s,t}$; coefficients $\alpha_t, \beta_t$; batch size $M$; diagonal fraction $\eta$;
weight $w_{s,t}$; sampling method $p_\gamma$.

**repeat**

> Sample $M_d = \lfloor \eta M \rfloor$ pairs $(x_0^i, x_1^i) \sim \rho(x_0, x_1)$;
> Sample $M_d$ times $t_i \sim U([0, 1])$;
> Compute interpolants $I_{t_i} = \alpha_{t_i} x_0^i + \beta_{t_i} x_1^i$ and velocities $\dot{I}_{t_i} = \dot{\alpha}_{t_i} x_0^i + \dot{\beta}_{t_i} x_1^i$;
> Compute diagonal loss: $\mathcal{L}_b = \frac{1}{M_d} \sum_{i=1}^{M_d} \left( e^{-w_{t_i,t_i}} |\hat{v}_{t_i,t_i}(I_{t_i}) - \dot{I}_{t_i}|^2 + w_{t_i,t_i} \right)$;
> Sample $M_o = M - M_d$ pairs $(x_0^j, x_1^j) \sim \rho(x_0, x_1)$;
> Sample $M_o$ pairs $(s_j, t_j) \sim U_{\mathsf{od}}$;
> Sample intermediate fractions: $\gamma_j \sim p_\gamma$ (e.g., $U([0, 1])$ or $\delta_{1/2}$);
> Compute intermediate times: $u_j = \gamma_j s_j + (1 - \gamma_j) t_j$;
> Compute interpolants: $I_{s_j} = \alpha_{s_j} x_0^j + \beta_{s_j} x_1^j$;
> Evaluate model at student points: $\hat{v}_{s_j, t_j}(I_{s_j})$;
> Evaluate model at first segment: $\hat{v}_{s_j, u_j}(I_{s_j})$;
> Compute intermediate flow maps: $\hat{X}_{s_j, u_j}(I_{s_j}) = I_{s_j} + (u_j - s_j)\hat{v}_{s_j, u_j}(I_{s_j})$;
> Evaluate model at second segment: $\hat{v}_{u_j, t_j}(\hat{X}_{s_j, u_j}(I_{s_j}))$;
> Compute preconditioned teacher signals:
> $\quad \hat{v}_{s_j, t_j}^{\text{teacher}} = (1 - \gamma_j)\hat{v}_{s_j, u_j}(I_{s_j}) + \gamma_j \hat{v}_{u_j, t_j}(\hat{X}_{s_j, u_j}(I_{s_j}))$;
> Compute residuals: $r_j = \hat{v}_{s_j, t_j}(I_{s_j}) - \mathsf{sg}\left( \hat{v}_{s_j, t_j}^{\text{teacher}} \right)$;
> Compute PSD loss: $\mathcal{L}_{\mathsf{PSD}} = \frac{1}{M_o} \sum_{j=1}^{M_o} \left( e^{-w_{s_j, t_j}} |r_j|^2 + w_{s_j, t_j} \right)$;
> Compute self-distillation loss: $\mathcal{L}_{\mathsf{SD}} = \mathcal{L}_b + \mathcal{L}_{\mathsf{PSD}}$;
> Update $\hat{v}$ and $w$ using $\nabla \mathcal{L}_{\mathsf{SD}}$;

**until** *converged*;

**output:** Trained flow map $\hat{X}_{s,t}(x) = x + (t - s)\hat{v}_{s,t}(x)$

---

## G   Further details on numerical experiments

Here, we provide a complete description of the numerical experiments performed in the main text. A concise summary of each experiment is given in Table 2.

### G.1   Checkerboard Details

**Experimental setup.**   We compare the LSD, ESD, and PSD algorithms on the two-dimensional checkerboard dataset. For PSD, we evaluate both uniform sampling ($\gamma \sim U([0, 1])$, denoted PSD-U) and midpoint sampling ($\gamma = 1/2$, denoted PSD-M). We generate a dataset with $10^7$ samples and train for $150,000$ steps with a batch size of $100,000$ and a learning rate of $10^{-3}$ with square root decay after $35,000$ steps. We use a diagonal fraction of $\eta = 0.75$, allocating 75% of each batch to the flow matching loss $\mathcal{L}_b$ and 25% to the self-distillation loss. The network architecture consists of a 4-layer MLP with $512$ neurons per hidden layer and GELU activation functions. We use the linear interpolant with $\alpha_t = 1 - t$ and $\beta_t = t$ with a Gaussian base distribution $x_0 \sim \mathsf{N}(0, I)$ with adaptive scaling to normalize by the variance of the target distribution. Times are sampled uniformly over the upper triangle without annealing, and we apply gradient clipping at 10.0. All methods use the stopgradient configurations described in (94). We visualize model samples produced from an exponential moving average of the learned parameters with decay factor 0.999. Each experiment was run on a single 40GB A100 GPU. A full qualitative visualization of the tabular results discussed in the main text is shown in Figure 6.

**KL Computation.**   To compute the KL divergence, we leverage that (a) the checkerboard density is known analytically as a uniform density over the selected squares, (b) the low-dimensionality of the dataset means that histogramming the model samples gives a good approximation of the model density, and (c) the low-dimensionality implies that quadrature can be used to compute a high-accuracy, deterministic approximation of KL. To this end, we first compute $64,000$ samples

| | Checker | CIFAR-10 | CelebA-64 | AFHQ-64 |
|---|---|---|---|---|
| *Dataset Properties* | | | | |
| Dimensionality | 2 | $3 \times 32 \times 32$ | $3 \times 64 \times 64$ | $3 \times 64 \times 64$ |
| Samples | $10^7$ | 50k | 203k | 16k |
| *Network* | | | | |
| Architecture | 4-layer MLP | EDM2 | EDM2 | EDM2 |
| Hidden/base channels | 512 | 128 | 128 | 128 |
| Channel multipliers | – | [2, 2, 2] | [1, 2, 3, 4] | [1, 2, 3, 4] |
| Residual blocks | – | 4 per resolution | 3 per resolution | 3 per resolution |
| Attention resolutions | – | $16 \times 16$ | $16 \times 16, 8 \times 8$ | $16 \times 16, 8 \times 8$ |
| Dropout | – | 0.13 | 0.0 | 0.0 |
| *Hyperparameters* | | | | |
| Batch size | 100,000 | 512 | 256 | 256 |
| Training steps | 150,000 | 400,000 | 800,000 | 800,000 |
| Total samples | $25 \times 10^9$ | $204.8 \times 10^6$ | $204.8 \times 10^6$ | $204.8 \times 10^6$ |
| Optimizer | RAdam | RAdam | RAdam | RAdam |
| Learning rate | $10^{-3}$ | $10^{-2}$ | $10^{-2}$ | $10^{-2}$ |
| LR schedule | Sqrt decay at 35k | Sqrt decay at 35k | Sqrt decay at 35k | Sqrt decay at 35k |
| Gradient clipping | 10.0 | 1.0 | 1.0 | 1.0 |
| Diagonal fraction $\eta$ | 0.75 | 0.75 | 0.75 | 0.75 |
| EMA decay | 0.999 | 0.9999 | 0.9999 | 0.9999 |
| *Evaluation* | | | | |
| Metric | KL divergence | FID | FID | FID |
| Sample count | 64,000 | 50,000 | 50,000 | 10,000 |
| *Methods* | | | | |
| Algorithms | LSD, ESD, PSD-U, PSD-M | LSD, PSD-U, PSD-M | LSD, PSD-U, PSD-M | LSD, PSD-U, PSD-M |

**Table 2: Experimental setup.** Summary of experimental configurations across all datasets. All experiments use uniform sampling of $(s, t)$ pairs over the upper triangle and leverage the sg $(\cdot)$ choices described in (94).

from each model for each number of steps $N$. We then histogram these samples using an $M \times M$ grid with $M = 50$ over the range $[-1, 1]^2$. To approximate the KL, we use the quadrature formula

$$\mathsf{KL}(\rho_1 \parallel \hat{\rho}_1) = \int \log \left( \frac{\rho_1(x)}{\hat{\rho}_1(x)} \right) \rho_1(x) dx \approx \sum_{i=1}^{M} \sum_{j=1}^{M} \log \left( \frac{\rho_1(x_{ij})}{\hat{\rho}_1^{\text{hist}}(x_{ij})} \right) \rho_1(x_{ij}) \Delta x \Delta y, \quad (103)$$

where in (103) $x_{ij}$ denotes the center of bin $(i, j)$ used to compute the histogram. We note that because of the uniformity of $\rho_1$ and $\hat{\rho}_1^{\text{hist}}$, this quadrature rule is exact, i.e.

$$\sum_{i=1}^{M} \sum_{j=1}^{M} \log \left( \frac{\rho_1(x_{ij})}{\hat{\rho}_1^{\text{hist}}(x_{ij})} \right) \rho_1(x_{ij}) \Delta x \Delta y = \mathsf{KL}(\rho_1 \parallel \hat{\rho}_1^{\text{hist}}). \quad (104)$$

### G.2 CIFAR-10 Details.

We evaluate the LSD, ESD, and PSD algorithms on the CIFAR-10 dataset. Again for PSD, we compare uniform sampling ($\gamma \sim U([0, 1])$, denoted PSD-U) and midpoint sampling ($\gamma = 1/2$, denoted PSD-M). All methods use uniform sampling over the upper triangle $(s, t) \sim U_{\text{od}}$ without annealing. We train for $400,000$ steps with a batch size of $512$ and an initial learning rate of $10^{-2}$ with square root decay after $35,000$ steps. We use a diagonal fraction of $\eta = 0.75$, allocating 75% of each batch to the flow matching loss and 25% to the self-distillation loss. The network architecture is based on EDM2 in Configuration G (Karras et al., 2024) and NCSN++ (Song et al., 2020), using 128 base channels, channel multipliers $[2, 2, 2]$, and 4 residual blocks per resolution. We use positional embeddings for time, as we found that Fourier embeddings led to greater training instability (Lu and Song, 2025). We embed $s$ and $(t - s)$ rather than $s$ and $t$, which we found to perform better in early experiments, add these embeddings together, and otherwise use standard FiLM conditioning in the EDM2 network. We apply attention at the $16 \times 16$ resolution and use dropout of $0.13$ following EDM

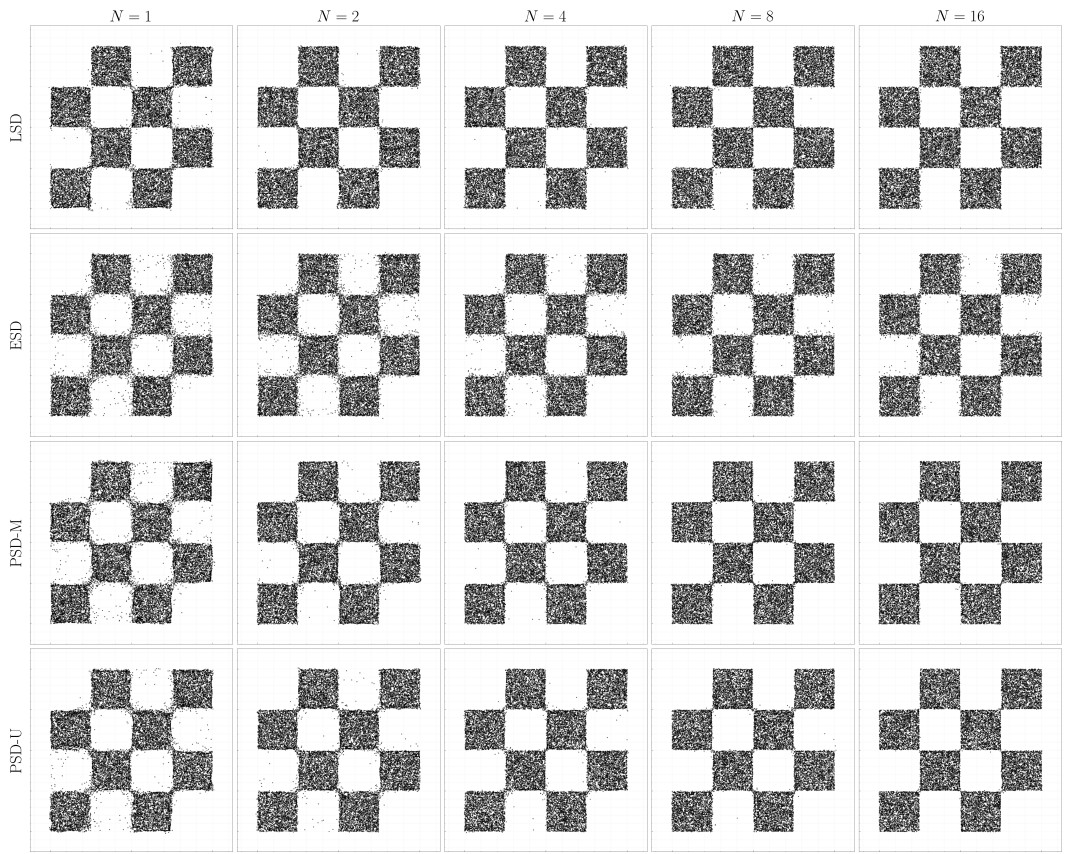

**Figure 6: Checker: full qualitative results** Full visualization of sample quality as a function of number of steps on the two-dimensional checker dataset.

recommendations for CIFAR-10 (Karras et al., 2022). We employ a learned weight function $w_{s,t}$ with 128 channels to normalize gradient variance. The interpolant uses $\alpha_t = 1 - t$ and $\beta_t = t$ with a Gaussian base distribution, setting the variance of the Gaussian adaptively to match the variance of the training data. We apply gradient clipping at 1.0 and use the stopgradient configurations described in (94) for LSD and PSD; ESD was unstable in every $\mathsf{sg}\,(\cdot)$ configuration tried. We evaluate sample quality using FID computed on-the-fly every $10,000$ steps with $10,000$ generated samples, using NFE $\in \{1, 2, 4, 8, 16\}$ for the flow map. Models were trained from random initialization without pre-training, and we track EMA parameters with decay factors $0.999$ and $0.9999$. We re-compute FID over $50,000$ generated samples for post-processing and take the best checkpoints for each number of sampling steps over the entire training range with an EMA factor $0.9999$. We use the RAdam optimizer with default settings. Minimal hyperparameter tuning was applied to the algorithms due to well-established training practices for CIFAR-10 available in the literature.

### G.3 CelebA-64 Details

We compare LSD and both PSD variants (uniform and midpoint) on the CelebA-64 dataset. As for CIFAR-10, we found ESD to be uniformly unstable and so do not report results. We train for $800,000$ steps (corresponding to $204.8$M samples) with a batch size of $256$ and an initial learning rate of $10^{-2}$ with square root decay after $35,000$ steps. We use a diagonal fraction of $\eta = 0.75$, allocating $75\%$ of each batch to the flow matching loss and $25\%$ to the self-distillation loss. The network architecture is based on EDM2 in Configuration G with 128 base channels, channel multipliers $[1, 2, 3, 4]$, and 3 residual blocks per resolution, corresponding to the "ImageNet-S" variant reduced from 192 channels to 128. We apply attention at resolutions $16 \times 16$ and $8 \times 8$, and do not use dropout. As with CIFAR-10, we use positional embeddings for time and embed $s$ and $(t - s)$ with standard FiLM conditioning. We use the linear interpolant with $\alpha_t = 1 - t$ and $\beta_t = t$ with a Gaussian base distribution and adaptive scaling to normalize to the variance of the target density. Times points $(s, t)$

are sampled uniformly over the upper triangle and no annealing or pretraining is used. We apply gradient clipping at $1.0$ and leverage the stopgradient configuration (94) for all methods. We use the RAdam optimizer with default settings. We evaluate online sample quality using FID-10K computed every $10,000$ steps, and then compute FID-50k *post-hoc* to find the best model, following the same steps as for CIFAR-10. Models were trained from random initialization without pre-training, and we track EMA parameters with decay factor $0.9999$. FID-50K scores were computed with this EMA factor.

## G.4   AFHQ-64 Details

We compare LSD and both PSD variants (uniform and midpoint) on the AFHQ-64 dataset. As with CelebA-64, we found ESD to be unstable and do not report results. We train for $800,000$ steps (corresponding to 204.8M samples) with a batch size of $256$ and an initial learning rate of $10^{-2}$ with square root decay after $35,000$ steps. We use a diagonal fraction of $\eta = 0.75$, allocating 75% of each batch to the flow matching loss and 25% to the self-distillation loss. The network architecture is based on EDM2 in Configuration G with 128 base channels, channel multipliers $[1, 2, 3, 4]$, and 3 residual blocks per resolution, matching the architecture used for CelebA-64. We apply attention at resolutions $16 \times 16$ and $8 \times 8$, and do not use dropout. As with CIFAR-10, we use positional embeddings for time and embed $s$ and $(t - s)$ with standard FiLM conditioning. We use the linear interpolant with $\alpha_t = 1 - t$ and $\beta_t = t$ with a Gaussian base distribution and adaptive scaling to normalize to the variance of the target density. Time points $(s, t)$ are sampled uniformly over the upper triangle and no annealing or pretraining is used. We apply gradient clipping at $1.0$ and leverage the stopgradient configuration (94) for all methods. We use the RAdam optimizer with default settings. We evaluate online sample quality using FID-10K computed every $10,000$ steps, and then compute FID-50k *post-hoc* to find the best model, following the same steps as for CIFAR-10 and CelebA-64. Models were trained from random initialization without pre-training, and we track EMA parameters with decay factor $0.9999$. FID-50K scores were computed with this EMA factor.

