# OpenReview forum: "How to build a consistency model: Learning flow maps via self-distillation"
_NeurIPS.cc/2025/Conference — NeurIPS 2025 poster_

### Official Review · Reviewer_1nz6 · 2025-07-02

**Clarity:** 4
**Significance:** 3
**Originality:** 2
**Rating:** 5
**Confidence:** 5

**Summary:**

The paper tackles the problem of training flow maps from scratch, without relying on a pretrained diffusion or flow-matching model. The authors begin with an analytical study of flow maps, deriving conditions under which a flow map is optimal. For each condition, they propose a corresponding training objective, each sharing the same global minimizer which is the optimal flow map. The proposed methods are evaluated on 2D toy datasets and the CIFAR-10 image generation benchmark, focusing primarily on the training dynamics and sample quality rather than achieving state-of-the-art performance.

**Questions:**

* Could the authors provide some more theoretical motivation or intuition for applying the stopgradient operation in lines 267-273?
* Besides computational resources, what are some of the challenges to scale up these techniques to more complicated datasets?
* Currently, most state-of-the-art diffusion models use a form of guidance (either classifier-free guidance or some of the newer alternatives such as autoguidance) during inference to achieve optimal results. Given this, many of the current distillation approaches[1,2] also distill the guidance scale into the student model. Is there a way of performing something similar when training flow maps from scratch?

**References**:

[1] Lu, C., & Song, Y. (2024). Simplifying, stabilizing and scaling continuous-time consistency models. arXiv preprint arXiv:2410.11081.

[2] Meng, C., Rombach, R., Gao, R., Kingma, D., Ermon, S., Ho, J., & Salimans, T. (2023). On distillation of guided diffusion models. In Proceedings of the IEEE/CVF Conference on Computer Vision and Pattern Recognition (pp. 14297-14306).

**Ethical Concerns:**

["NO or VERY MINOR ethics concerns only"]

**Final Justification:**

The paper is a well written one about training few-step flowmap models. The experiments section were a bit lacking (namely toy checkerboard data and CIFAR10) which has been partially addressed by adding additional CelebA-64 experiments, with ImageNet256 ones to follow for the final version. Conditioned on the presence of the ImageNet256 experiments in the final paper, I raised my rating from $4\to5$ and recommend accepting this paper.

**Limitations:**

Yes.

**Paper Formatting Concerns:**

No concerns.

**Quality:**

4

**Strengths And Weaknesses:**

**Strengths:**
* The paper is very well written and easy to follow.
* The derived algorithms and objectives are clearly motivated and presented.
* The proposed framework unifies several existing approaches, including consistency models and shortcut models.
* The paper offers new insights and intuitions into structure, behavior, and training dynamics of flow maps.

**Weaknesses:**
* The main limitation of this work is the narrow scope of the experimental evaluation. While the paper’s focus is primarily theoretical, the practical results are somewhat underwhelming. For example, the ESD approach, which is closely related in spirit to the state-of-the-art consistency model sCM [1], fails to converge in the reported experiments. Why does this approach fail where sCM succeeds? Comparing ESD, LSD, and PSD on toy problems may not reflect their true potential, especially when architectural choices are not carefully optimized. A more thorough empirical comparison under stronger settings could yield very different conclusions.
* I appreciate the discussion on the choice of teacher (lines 267-273), but the motivation for applying a stop-gradient remains unclear. Unlike the rest of the paper, which is grounded in clear theory, the use of stop-gradient feels arbitrary and ad hoc. Is its application dependent on the dataset? The model architecture? Does it depend on the data modality or dimensionality? Or something else? Furthermore, it would be helpful to clarify whether applying stop-gradient has any implications for the theoretical arguments made in Proposition 2.2.

**References**:

[1] Lu, C., & Song, Y. (2024). Simplifying, stabilizing and scaling continuous-time consistency models. arXiv preprint arXiv:2410.11081.

---

> ### Author Rebuttal · Authors · 2025-07-31
>
> We thank the reviewer for their careful reading and evaluation of our work. Below, we address the main questions and weaknesses raised. If our arguments are found convincing, we would greatly appreciate the reviewer raising their score. We would equally appreciate further suggestions for us to improve our work, and will happily provide further clarification on any aspects of our paper or our answers.
>
> ## Weaknesses
> *The main limitation of this work is the narrow scope of the experimental evaluation. While the paper’s focus is primarily theoretical, the practical results are somewhat underwhelming. For example, the ESD approach, which is closely related in spirit to the state-of-the-art consistency model sCM [1], fails to converge in the reported experiments. Why does this approach fail where sCM succeeds? Comparing ESD, LSD, and PSD on toy problems may not reflect their true potential, especially when architectural choices are not carefully optimized. A more thorough empirical comparison under stronger settings could yield very different conclusions.*
>
> We thank the reviewer for bringing up this important point, and we agree that more experiments would strengthen the contribution. To account for this, we have run a round of experiments on the CelebA-64 dataset, for which FID results are reported below.
>
>
> | Method | 1 step | 2 step | 4 step | 8 step | 16 step |
> |--------|--------|--------|--------|--------|---------|
> |   LSD    | **15.77**       | **8.53**       |  **4.70**      |  **3.20**      | **2.66**        |
> |   PSD-M     |  17.24      |   9.34     |   5.77     |  3.97      |  3.25       |
> |   PSD-U     |  17.94      |  9.19     |   5.55     |   4.11    |   3.55      |
>
>
> The above results are similar to what we saw on CIFAR-10, with ESD demonstrating instability and LSD and PSD performing well. Unlike on CIFAR-10, in this case, we find that LSD performs the best, supporting our hypothesis that the optimal training scheme is dataset and architecture-dependent. In addition to these new results on CIFAR-10, we are currently running a round of experiments using the SiT architecture on ImageNet256 [1], which will be completed in time for the camera ready version.
>
> **Connection to consistency models.** We agree with the reviewer that ESD is similar to consistency models. However, many recent works have highlighted difficulties with training consistency models in continuous time [2,3], which is very similar to what we observe. While these instabilities can be avoided with clever, hand-designed normalization schemes and architectural modifications, our aim here is to study the performance of our three introduced training schemes with minimal engineering. We anticipate that all algorithms can be engineered significantly to maximize performance, and believe it is a very interesting direction of future work to incorporate these recent stabilization schemes into ESD, LSD, and PSD. Nevertheless, we find it particularly intriguing that LSD and PSD are stable without these engineering efforts, which suggests that they may be preferred in practice.
>
> *I appreciate the discussion on the choice of teacher (lines 267-273), but the motivation for applying a stop-gradient remains unclear. Unlike the rest of the paper, which is grounded in clear theory, the use of stop-gradient feels arbitrary and ad hoc. Is its application dependent on the dataset? The model architecture? Does it depend on the data modality or dimensionality? Or something else? Furthermore, it would be helpful to clarify whether applying stop-gradient has any implications for the theoretical arguments made in Proposition 2.2.*
>
> Our algorithms are based on the concept of self-distillation, which converts a distillation scheme into a direct training scheme by indentifying an implicit source of signal. Our aim in this section is to draw an analogy to more standard distillation schemes, in which there is a frozen teacher model that is converted into a more performant model. From this perspective, it is intuitive that it may be beneficial to leverage a stopgradient operator to "freeze" the implicit teacher when optimizing the student. Mathematically, one way to motivate this is that the objective functions we introduce are nonconvex in the model $\hat{v}$ in a functional sense -- even before the parametric nonlinearity introduced by neural networks. This means that there may be spurious local minima, and optimizing the objective without using a stopgradient operator could reduce its value by changing the teacher to match the student, rather than vice-versa. In the revised paper, we significantly expanded this section to make these points more clear. Moreover, we have included some ablation studies over the choice of the stopgradient operator for LSD, PSD, and ESD on the checkerboard, CIFAR-10, and CelebA-64 datasets. In general, we find that using intuition from the distillation setting and applying a stopgradient to freeze the implicit teacher leads to the best results, though we do expect these conclusions to depend at least somewhat on the dataset and architecture. Our recommendation would be to ablate over these choices for a new dataset or problem domain to determine the right choice. We emphasize that this does not affect our theoretical arguments, which study properties of the loss value and the minimizers of the loss, The choice of stopgradient only affects the optimization dynamics.
>
> We agree with the reviewer that stopgradients are somewhat *ad-hoc*, and we believe that making this intuition more rigorous, as well as understanding the properties of the stopgradient from a mathematical level, is a very interesting direction of future research.
>
> ## Questions
> *Besides computational resources, what are some of the challenges to scale up these techniques to more complicated datasets?*
>
> The associated issues are mostly computational in nature, due to the combinatorial number of ablations needed to study each algorithm systematically. We are particularly interested, in future work, to leverage recent stabilization schemes for training continuous-time consistency models [2,3].
>
> *Currently, most state-of-the-art diffusion models use a form of guidance (either classifier-free guidance or some of the newer alternatives such as autoguidance) during inference to achieve optimal results. Given this, many of the current distillation approaches [1,2] also distill the guidance scale into the student model. Is there a way of performing something similar when training flow maps from scratch?*
>
> We thank the reviewer for this excellent suggestion. In general, our flow maps can be made conditional by incorporating a conditioning argument $$X\_{s, t}(x; c) = x + (t-s)\hat{v}\_{s, t}(x;c)$$ for a conditioning variable $c$. The flow $v\_{t, t}(x; c)$ then corresponds to a conditional velocity field. Given this observation, it is clear how we can defined guided flow maps, as we now describe.
>
> Let $c = \emptyset$ correspond to unconditional generation, and let $$q\_t(x; \alpha, c) = b\_t(x; \emptyset) + \alpha(b\_t(x; c) - b\_t(x; \emptyset))$$ correspond to the guided velocity field at guidance strength $\alpha$. This velocity field has a flow map $$X\_{s, t}(x; \alpha, c) = x + (t-s)v\_{s, t}(x; \alpha, c)$$ satisfying $v\_{t, t}(x; \alpha, c) = q\_t(x; \alpha, c)$, which may be learned via self-distillation using any of our approaches by incorporating additional random sampling over the guidance scale in the self-distillation term. We have added a paragraph titled "Classifier-free guidance" under the "Algorithmic aspects" section of our revised paper describing this approach.
>
>
> ## References
> [1] Ma et al, "SiT: Exploring Flow and Diffusion-based Generative Models with Scalable Interpolant Transformers", arXiv:2401.08740.
>
> [2] Le et al, "Simplifying, Stabilizing and Scaling Continuous-Time Consistency Models", arXiv:2410.11081.
>
> [3] Sabour et al, "Align Your Flow: Scaling Continuous-Time Flow Map Distillation", arXiv:2506.14603.

---

> > ### Comment · Reviewer_1nz6 · 2025-08-06
> >
> > I appreciate the authors' response, in particular the additional experiments and clarifications. I'm happy to see the experiments improved by including CelebA-64 (and potentially ImageNet256) experiments. It's also quite intriguing how the ESD approach seems to fail to train without additional normalization techniques, but the PSD/LSD approaches do not have this problem. My main concerns have been addressed and I recommend accepting this paper.

---

### Official Review · Reviewer_ySYP · 2025-07-03

**Clarity:** 3
**Significance:** 3
**Originality:** 3
**Rating:** 5
**Confidence:** 3

**Summary:**

This paper introduces a general framework for distilling ODE-based generative models. The core proposal is a "flow map" framework that unifies different distillation schemes, including consistency models and progressive flow map matching (PFMM), under a single theoretical framework. The authors show that this flow map is characterized by three key conditions (Lagrangian, Eulerian, and Semigroup) which correspond to different distillation objectives.

**Questions:**

Please check the weakness part.

**Ethical Concerns:**

["NO or VERY MINOR ethics concerns only"]

**Final Justification:**

I thank the authors for their detailed and comprehensive rebuttal, which has successfully addressed all of my major concerns.
The authors clarified the relationship between their framework and existing methods. They also provided new, promising results on CelebA-64 and will include large-scale experiments on ImageNet-256 in the final version.
I have raised my score to reflect these improvements.

**Limitations:**

Yes

**Quality:**

3

**Strengths And Weaknesses:**

+ The theoretical work is solid, providing a clean, unifying perspective on various distillation methods from an ODE standpoint.
+ The framework itself is elegant and helps clarify the relationships between different techniques.
+ The explanation of the core methodology is clear and well-written, making the theoretical contributions easy to follow.


- Missing Proof of Proposition 2.2: The proposition is central to the paper's theory, stating that a map satisfying certain equations becomes a valid flow map. However, a proof for this claim appears to be missing. Could the authors include a proof or point to one in the literature?
- Clarity on Novelty vs. Existing Work: The paper states that the Eulerian (ESD) and Semigroup (PSD) conditions correspond to consistency models and progressive distillation, respectively. This raises a key question: are the training objectives used for ESD and PSD identical to these existing methods? A more explicit comparison of the loss functions would help clarify the precise contribution and novelty of the proposed distillation schemes.
- Scope of Experiments: The experiments are limited to a toy dataset and CIFAR-10. To convincingly demonstrate the effectiveness and scalability of the proposed framework, it would be beneficial to include results on more complex, higher-resolution datasets (e.g., CelebA-64, LSUN or ImageNet-64).
- Ablation Studies: The paper proposes several training strategies, including different loss formulations (e.g., a linear combination of objectives, as mentioned in Proposition 2.3) and sampling strategies. However, the ablation studies only seem to cover the sampling strategy.
- Writing: The connection between the proposed framework and existing methods like consistency models is a major strength. Introducing these connections earlier in the main text, rather than later, would help readers better situate the work and appreciate its relationship to the current literature from the start.

---

> ### Author Rebuttal · Authors · 2025-07-31
>
> We thank the reviewer for their careful reading and evaluation of our work. Below, we address the main questions and weaknesses raised. If our arguments are found convincing, we would greatly appreciate the reviewer raising their score. We would equally appreciate further suggestions for us to improve our work, and will happily provide further clarification on any aspects of our paper or our answers.
>
> ## Weaknesses
> *Missing Proof of Proposition 2.2: The proposition is central to the paper's theory, stating that a map satisfying certain equations becomes a valid flow map. However, a proof for this claim appears to be missing. Could the authors include a proof or point to one in the literature?*
>
> We thank the reviewer for pointing out this organizational aspect of the original submission. The proof of Proposition 2.2 was split between the proofs of Propositions C.2, C.3, and C.4 in the appendix, which break down Proposition 2.2 into smaller results. We agree that this may have made it challenging to quickly reference the proof, and that this structure was not properly indicated in the text. Following the reviewer's advice, we have organized the proofs in the appendix so that Proposition 2.2 is proven in a self-contained fashion, and so that Propositions C.2, C.3, and C.4 continue to stand on their own in proving further theoretical guarantees.
>
> *Clarity on Novelty vs. Existing Work: The paper states that the Eulerian (ESD) and Semigroup (PSD) conditions correspond to consistency models and progressive distillation, respectively. This raises a key question: are the training objectives used for ESD and PSD identical to these existing methods? A more explicit comparison of the loss functions would help clarify the precise contribution and novelty of the proposed distillation schemes.*
>
> When we use the language "corresponds to", we mean that these methods are based on the underlying properties that we identify (the Eulerian equation or the semigroup condition). This is a novel contribution in its own right, as these past methods were motivated in a more *ad-hoc* fashion. Nevertheless, our contributions go beyond this additional level of theoretical clarity, and there are several key differences between consistency training, progressive distillation, and the approaches that we introduce here. In particular:
>
> 1. Our methodological framework is valid for two-time flow maps defined over general stochastic interpolants, while consistency models are single-time flow maps over diffusion processes. Restricting to this setting -- i.e., choosing the interpolant to correspond to a diffusion process, choosing to estimate only the noise to data map $f_s(x) = X_{s, 1}(x)$, and using Eulerian distillation on a pre-trained model -- recovers consistency distillation [1,2].
> 1. Eulerian self-distillation is new, but is related to consistency training. In consistency training, the loss function corresponds to the squared Eulerian residual where the target flow model $b$ is replaced by the derivative of the interpolant $\dot{I}\_s$. Mathematically, $\mathcal{L}_{\mathsf{CT}}(\hat{f}) = \int\_0^1 \mathbb{E}\left[|\partial\_s \hat{f}\_s(I\_s) + \nabla \hat{f}\_s(I\_s)\dot{I}\_s|^2\right]ds$. Because $b\_s(x) = \mathbb{E}[\dot{I}\_s | I\_s = x]$, and because the consistency training objective does not incorporate this conditioning information, it introduces an uncontrolled bias that needs to be sidestepped with complex training schemes [3]. Our approach can be viewed as a more principled way to train that leverages the implicit flow defined by the tangent condition $\hat{v}\_{s, s}(I\_s)$ instead of the derivative of the interpolant $\dot{I}\_s$.
> 1. Progressive distillation [4] is most related to the progressive flow map matching algorithm introduced in [2], which systematically converts a given flow map into a hierarchy of flow maps that can take larger steps, starting from an original velocity model. Critically, each stage requires a new neural network that learns to emulate the previous one. The PSD approach we introduce is similar, but can be viewed as a continuous variant that does not require multiple models. As a result, the objective function is different and is much simpler to use.
>
> To make these points more clear, we have added a new section to the appendix "Connections to existing schemes" that significantly clarifies the relation between our approaches and these pre-existing methods. In particular, we include two subsections -- "Distillation" and "Direct training" -- which highlight how our framework reduces to, or is related to, existing methods in specific instantiations (consistency distillation, progressive distillation, shortcut models, consistency trajectory models, etc.). Our methodology is the most general framework for training consistency models that we are aware of, and one of our primary contributions is that we are the first authors to draw rigorous connections between these existing approaches from a single common viewpoint.
>
>
> **Scope of Experiments: The experiments are limited to a toy dataset and CIFAR-10. To convincingly demonstrate the effectiveness and scalability of the proposed framework, it would be beneficial to include results on more complex, higher-resolution datasets (e.g., CelebA-64, LSUN or ImageNet-64).**
>
> We thank the reviewer for raising this important point. Since receiving the reviews, we have completed a round of experiments on CelebA-64, which support our hypothesis that the optimal training scheme is dataset and network-dependent. Here, similar to CIFAR-10, we find that ESD is unstable, which we hypothesize originates from the spatial Jacobian of the UNet. PSD and LSD both perform well. Unlike CIFAR-10, here we find that LSD performs the best, matching our results from the checkerboard. Numerical FID values included in an updated version of the text are shown below.
>
> | Method | 1 step | 2 step | 4 step | 8 step | 16 step |
> |--------|--------|--------|--------|--------|---------|
> |   LSD    | **15.77**       | **8.53**       |  **4.70**      |  **3.20**      | **2.66**        |
> |   PSD-M     |  17.24      |   9.34     |   5.77     |  3.97      |  3.25       |
> |   PSD-U     |  17.94      |  9.19     |   5.55     |   4.11    |   3.55      |
>
>
> We are currrently running another round of large-scale experiments using the SiT architecture [5] on ImageNet-256, which we intend to have completed for the camera-ready version.
>
> *Ablation Studies: The paper proposes several training strategies, including different loss formulations (e.g., a linear combination of objectives, as mentioned in Proposition 2.3) and sampling strategies. However, the ablation studies only seem to cover the sampling strategy.*
>
> We agree with the reviewer that there are several suggestions made in the text -- such as combinations of objectives, multiple models, and higher-order parameterizations -- that we did not ablate over. In this work, our aim is to introduce a systematic mathematical framework that analytically characterizes the design space of consistency models. Due to the combinatorial volume and associated computational expense of ablation studies over all factors, we are unable to include experiments testing every combination. Instead, we have studied a representative subset, including the three key choices of the objective and the sampling strategy for PSD. In an updated revision, we have also included some experiments studying the role of the stopgrad operator, where we highlight that placing a stopgrad to make the loss convex in the student typically produces the best results in high-dimensional settings. By laying out the design space completely as we have done here, researchers who wish to deploy a consistency model in their application domain of interest will have a guidebook for the hyperparameters they can study to maximize performance.
>
> *Writing: The connection between the proposed framework and existing methods like consistency models is a major strength. Introducing these connections earlier in the main text, rather than later, would help readers better situate the work and appreciate its relationship to the current literature from the start.*
>
> We agree with the reviewer that a strength of our contribution is its connection to existing methods, and we thank them for noticing this. In an updated revision, we have included further details on these connections up front. Moreover, we have added an additional appendix that includes all necessary derivations to map our approach onto existing methodologies.
>
>
> ## References
> [1] Song et al, “Consistency Models.” arXiv:2303.01469.
>
> [2] Boffi et al, "Flow Map Matching with Stochastic Interpolants: A Mathematical Framework for Consistency Models." arXiv:2406.07507.
>
> [3] Lu et al, “Simplifying, Stabilizing and Scaling Continuous-Time Consistency Models.” arXiv:2410.11081.
>
> [4] Salimans et al, "Progressive Distillation for Fast Sampling of Diffusion Models." arXiv:2202.00512.
>
> [5] Ma et al, "SiT: Exploring Flow and Diffusion-based Generative Models with Scalable Interpolant Transformers", arXiv:2401.08740.

---

> > ### Comment · Reviewer_ySYP · 2025-08-06
> > **Reply**
> >
> > I thank the authors for their detailed rebuttal which has greatly addressed my concerns, and I look forward to the inclusion of the CelebA and ImageNet results which will make the paper much stronger experimentally.

---

### Official Review · Reviewer_2Efm · 2025-07-08

**Clarity:** 3
**Significance:** 3
**Originality:** 3
**Rating:** 4
**Confidence:** 3

**Summary:**

The manuscript focuses on learning flow map-based models and how they can be converted into self-distillation models. One major advantage of using flow map models over traditional flow models is the inference-time speedup, as they do not require ODE or SDE discretization. Most existing flow map-based methods follow a two-stage setup: first learning a flow (teacher), then learning its solution (student). In contrast, this manuscript centers on directly learning the flow map.

**Questions:**

Line 133: "along trajectories of the probability flow" — This is slightly misleading, as it does not follow the full trajectory. Rather, it only requires correct prediction of $X_t$ given $X_s$ (as also illustrated in Figure 1b).

Proposition 2.4 defines an $\epsilon$ with respect to ESD, but the remark on line 201 references both ESD and LSD. Could you define an $\epsilon$ for LSD as well?

The notation in Equation (17) is somewhat confusing, particularly the first exponential term.

In proposition 2.4, L is the constant for $v_{t,t}$, is it for all t?  For example, by taking the sup over all t ?

**Ethical Concerns:**

["NO or VERY MINOR ethics concerns only"]

**Final Justification:**

I would like to thank the authors for the fruitful discussion. I will keep my score as it is. In my opinion, the experimental section is insufficient. The experiments primarily report FID, yet it is also argued that too much importance should not be placed on FID values. The experimental section should address one key question: How accurate are the methods at learning flow maps? With the current setup, this question is only partially answered by showing FID for different step counts. Alternative experimental designs could have better addressed this, for example, by reporting FID scores for the same step counts using a model not trained with the self-distillation objective.

**Limitations:**

The limitations of the approach should be discussed in a concluding section or highlighted through experiments.

**Paper Formatting Concerns:**

No concerns.

**Quality:**

2

**Strengths And Weaknesses:**

Strengths

- The manuscript is very well written and clear.

- The introduction and background sections are concise yet informative.

- The method is theoretically grounded. Under certain assumptions, the authors derive a bound on the approximation error (in terms of 2-Wasserstein distance), which depends on a Lipschitz constant. They also directly relate the error with the training loss.

Weaknesses

My understanding is the approximation quality is inherently limited by how well a truncated Taylor polynomial can approximate the flow map. There is no a priori guarantee that the Taylor approximation will perform well.

The error bound depends on the Lipschitz constant. One way to achieve a smaller error is by inducing “straighter” flows. A known strategy for this is to use optimal transport (OT), or mini-batch OT, in flow matching. In that setting, the training remains the same, but the pairs $(x_0,x_1)$ are drawn from the OT coupling [1]. This could also be applied here, where in Algorithm 1, the authors sample $(x_0,x_1)$ from the OT coupling. Trying this might help alleviate the lack of control over the quality of the Taylor approximation.

Overall, I find the experimental section insufficient, particularly due to the lack of comparisons. While I agree that achieving state-of-the-art performance is not necessary, at a minimum, the authors should include results for a naive two-stage self-distillation baseline. Even if the results are similar, the proposed method remains valuable because it only requires one training phase.

[1] Tong, Alexander, et al. Improving and generalizing flow-based generative models with minibatch optimal transport. arXiv preprint arXiv:2302.00482 (2023).

---

> ### Author Rebuttal · Authors · 2025-07-31
>
> We thank the reviewer for their careful reading and evaluation of our work. Below, we address the main questions and weaknesses raised. If our arguments are found convincing, we would greatly appreciate the reviewer raising their score. We would equally appreciate further suggestions for us to improve our work, and will happily provide further clarification on any aspects of our paper or our answers.
>
> ## Weaknesses
> *My understanding is the approximation quality is inherently limited by how well a truncated Taylor polynomial can approximate the flow map. There is no a priori guarantee that the Taylor approximation will perform well.*
>
> We thank the reviwer for this important question, which was also raised by Reviewer cfDL. While the parameterization that we exploit is reminiscent of an Euler step, and while we find it useful to think about it in this way, we emphasize that the ansatz $X_{s, t}(x) = x + (t-s)v_{s, t}(x)$ is without loss of generality and introduces no approximation error. To see this, we may observe that the function $v_{s, t}(x) = \frac{1}{t-s}(X_{s, t}(x) - x)$ is simply a time-dependent shift and rescaling of the flow map.
>
> Our rationale for choosing this representation is computational: it eliminates the need to compute the limiting partial derivative $\lim_{s\to t}\partial_t X_{s, t}(x)$ that appears in the tangent identity, which our approach exploits to derive an off-diagonal signal. As we show in the text, this limit is precisely given by $v_{t, t}(x)$, and hence reduces to a single neural network evaluation rather than a Jacobian-vector product. In the revision, we have added several lines of clarification around the introduction of this parameterization. In particular, we emphasize that there is no approximation or limitation in using it.
>
> We would also like to emphasize that our algorithmic formulation is general and allows for an arbitrary parameterization of the map, though this choice is a natural one for the reasons just described.
>
> *The error bound depends on the Lipschitz constant. One way to achieve a smaller error is by inducing “straighter” flows. A known strategy for this is to use optimal transport (OT), or mini-batch OT, in flow matching. In that setting, the training remains the same, but the pairs  are drawn from the OT coupling [1]. This could also be applied here, where in Algorithm 1, the authors sample  from the OT coupling. Trying this might help alleviate the lack of control over the quality of the Taylor approximation.*
>
> We thank the reviewer for this very interesting suggestion. In response, we have added a short discussion of the role of couplings (and in particular mini-batch OT) in the "Algorithmic aspects" section, emphasizing that they made significantly improve the training. Given the large amount of contributions already present in the paper, we feel that a thorough investigation of the role of couplings would be best left for future work.
>
> *Overall, I find the experimental section insufficient, particularly due to the lack of comparisons. While I agree that achieving state-of-the-art performance is not necessary, at a minimum, the authors should include results for a naive two-stage self-distillation baseline. Even if the results are similar, the proposed method remains valuable because it only requires one training phase.**
>
> We thank the reviewer for suggesting this important baseline, as well as for suggesting that we improve the quality of the experimental section. To address this concern, we have completed an additional set of runs on the CelebA-64 dataset, which we feel significantly expands the scope of the present work.
>
> We agree that a comparison to a distillation baseline could be a natural experiment to include. However, we would like to respectfully point out that it is somewhat unclear what this baseline should be. We are not aware of a distillation baseline for PSD outside of progressive distillation [1], which requires recursively training a hierarchy of models and as a result is a much more complicated pipeline than what we propose here. We show that ESD is unstable in comparison to PSD and LSD, which was also observed in prior work when used for distillation on image data, making a baseline comparison challenging [2]. While clever stopgrad-based tricks have recently been introduced in the literature to stabilize Eulerian methods [3], incorporating these approaches into our paradigm is out of scope for the present work: in particular, we expect similar techniques can be used to improve PSD and LSD, which we plan to investigate systematically in future projects.
>
> These observations leave only the LSD algorithm, for which a natural distillation baseline would be the Lagrangian Map Distillation (LMD) approach [2]. If the reviewer would like to see the associated FID scores for comparison, we will gladly include them in the camera ready version. In the meantime, because of this difficulty with defining an appropriate baseline, we have chosen to focus on the new round of experiments on CelebA-64.
>
> *The limitations of the approach should be discussed in a concluding section or highlighted through experiments.*
>
> The main limitation of our present contribution is that, while we analytically characterize the vast design space of consistency models, we are unable to systematically test each component empirically due to the large associated computational expense. Critical aspects of the pipeline that deserve further testing include ablations over the flow map parameterization, stabilization and stopgradient schemes, as well as annealing schemes. In the camera-ready version, we will include a section covering this important limitation.
>
> ## Questions
> *Line 133: "along trajectories of the probability flow" — This is slightly misleading, as it does not follow the full trajectory. Rather, it only requires correct prediction of $X_t$ given $X_s$ (as also illustrated in Figure 1b).*
>
> We agree with the reviewer that the flow map can take large jumps along trajectories, and hence it does not necessarily follow the full trajectory. What we mean by this statement is that the flow map always maps along a single trajectory by definition -- i.e., $X_s$ and $X_t$ lie on the same trajectory. In particular, in the limit of an infinite number of steps, we precisely recover a trajectory of the underlying flow. We will happily clarify this in the camera-ready version.
>
> *Proposition 2.4 defines an $\epsilon$ with respect to ESD, but the remark on line 201 references both ESD and LSD. Could you define an  $\epsilon$ for LSD as well?*
>
> We agree with the reviewer's observation that the current statement of the theorem may be misleading. In the camera-ready version, we will provide an updated version of the theorem that clearly states the assumptions that $\mathcal{L}_b + \mathcal{L}\_{\mathsf{ESD}} \leq \epsilon$ or $\mathcal{L}_b + \mathcal{L}\_{\mathsf{LSD}} \leq \epsilon$.
>
> *The notation in Equation (17) is somewhat confusing, particularly the first exponential term.*
>
> The expression is $2 e\times (1 + e^{2\hat{L}})$ -- i.e., there is no first exponential term, but it is the constant $e$. We agree that as-written this expression may be misleading and we will clarify it in the revision.
>
> *In proposition 2.4, $L$ is the constant for $\hat{v}_{t, t}$, is it for all t? For example, by taking the sup over all $t$?*
>
> Yes, we assume that $\hat{v}_{t, t}$ is uniformly Lipschitz in time $t$. We will make this statement more clear in the revision.
>
>
> ## References
> [1] Salimans et al, "Progressive Distillation for Fast Sampling of Diffusion Models." arXiv:2202.00512.
>
> [2] Boffi et al, "Flow Map Matching with Stochastic Interpolants: A Mathematical Framework for Consistency Models." arXiv:2406.07507.
>
> [3] Sabour et al, "Align Your Flow: Scaling Continuous-Time Flow Map Distillation", arXiv:2506.14603.

---

> > ### Comment · Reviewer_2Efm · 2025-08-06
> >
> > Thank you for answering my questions and for providing clarifications and additional experiments. I believe that incorporating LSD distillation would strengthen the experimental section.
> >
> > I have one other comment. While I understand that this specific ansatz does not introduce any error, I believe it would be valuable to empirically demonstrate how the error depends on the Lipschitz constant. A toy example in which the constant can be varied would serve this purpose well. This would nicely complement the theoretical results (e.g., Proposition 2.4).

---

> > > ### Author Response · Authors · 2025-08-07
> > >
> > > Thanks for your reply. We will gladly include a Lagrangian distillation baseline in the camera-ready version.
> > >
> > > We agree with your suggestion about studying the effect of the Lipschitz constant. Similarly, we will happily include a Gaussian mixture example where the dimensionality and Lipschitz constant can be tuned systematically.

---

> > > > ### Comment · Reviewer_2Efm · 2025-08-07
> > > >
> > > > Thank you for adding these! Related to this, I wonder if you compute the Lipschitz constant of the vector field network (or an approximation thereof) for all methods in Table 2, would there be a relationship with the FID results? Could this help explain why LSD performs poorly, and why all methods require a larger step count to achieve decent results?

---

> > > > > ### Author Response · Authors · 2025-08-08
> > > > >
> > > > > We thank the reviewer for this thoughtful and intriguing suggestion, and we agree that it would be an excellent empirical addition to support our theoretical Wasserstein bounds. To assess the associated feasibility, we performed a literature review on methods for computing Lipschitz constants of neural networks. We found that, while there has been interest in their computation (particularly in the context of adversarial examples and in reinforcement learning-based control), existing algorithms are complex and known to produce coarse upper bounds [1, 2, 3]. For this reason, we feel that we do not have time to perform this calculation before the end of the discussion period, and that it would be best left to future work.
> > > > > Please let us know if there is anything else we can answer or discuss about the manuscript.
> > > > >
> > > > > **References**
> > > > >
> > > > > [1] Efficient and Accurate Estimation of Lipschitz Constants for Deep Neural Networks. Fazlyab et. al. (2019).
> > > > >
> > > > > [2] Lipschitz regularity of deep neural networks: analysis and efficient estimation. Scaman et. al. (2018).
> > > > >
> > > > > [3] Efficiently Computing Local Lipschitz Constants of Neural Networks via Bound Propagation. Shi et. al. (2022).

---

### Official Review · Reviewer_cfDL · 2025-07-11

**Clarity:** 3
**Significance:** 2
**Originality:** 2
**Rating:** 4
**Confidence:** 5

**Summary:**

This paper proposes a two-time flow-matching approach to generative modelling, closely related to diffusion and flow matching. Instead of learning only a vector field (that acts in the continuity equation), the paper proposes learning a two-time flow map (that acts in the pushforward equation), which, in theory, allows one to map directly from noise to data in one function call. The paper models this flow map with a Euler-step discretisation of a vector field model.

Based on the theoretical properties of the flow map, the paper proposes a loss function (Eq. 11) that decomposes into a standard flow matching loss (Eq. 12) and a combination of three other derived losses (Eq. 13-15) – LSD (matches final time derivative to vector field), ESD (matches the initial time derivative to the vector field), and PSD (matches the semigroup property of the flow map). The paper then devises four different models – LSD, ESD, PSD-M, and PSD-U – whose loss function corresponds to the three derived losses (with PSD-{U, M} being sub-variants).

The paper then tests them on the Checkers dataset (in terms of KL-divergence) and CIFAR-10 dataset (in terms of FID). Results show that in the Checkers dataset (lower-dimensional, higher batch size) that the PSD approach wins for higher step sizes, while for the CIFAR-10 dataset (higher-dimensional, lower batch size) that the PSD approach wins for lower step sizes.

**Questions:**

1. Can you please justify the first-order (Euler) flow-map parameterization (Eq. 6)?
    - Why is it sufficient, and were higher-order forms tried?
    - Can you provide an ablation comparing Euler vs. a 2-step Heun/midpoint alternative, or provide a theoretical argument that higher-order terms offer no practical gain?
    - This assumption seems to underpin every experiment – if it is a hidden bottleneck, results might not generalise.
2. Can you report the compute-performance trade-off of $\eta$-sampling (mentioned in line 243)?
    - Can you provide an experiment that sweeps $\eta$ and plots validation loss/FID vs. training cost?
3. Can you evaluate the proposed "Multiple-model" and "General representation" variants (mentioned in lines 209 and 218)?
    - Providing at least a small ablation (e.g., on MNIST) would show whether the single-network design is a conscious trade-off.
    - If impractical, can you provide guidance on what the effects of representations would be and when to prefer them?
    - These variants are positioned as advantages, but were not validated.
4. Can you provide more details about the KL divergences in Table 1?
    - Can you specify how the KL is estimated, which subset of data is held out, and report standard error over at least 3 runs?
5. Can you address the fairness of the comparison between Checker (Table 1) and CIFAR-10 (Table 2)?
    - Why is it that the trends in Tables 1 and 2 seem to be opposite? Vector field supervision appears to be more effective for small steps in Checkers, whereas flow map only supervision is more effective for small steps in CIFAR-10, and vice versa.
    - How do these trends relate to the data dimension? Could you do an experiment that scales dimension and/or batch size?
    - How fair is comparing these trends, given that the batch size in Checkers is 64000 and in CIFAR-10 is 512?
    - In Table 2, why does PSD-M get better than PSD-U for more steps? Also, why does this trend not seem to show in Table 1?
6. Can you elaborate on why we do not also train on the lower diagonal of the unit square (ref Fig. 2) for reverse steps (data to noise direction)?
    - What would be the effect of doing so?
6. Miscellaneous:
    - Can you elaborate on what "Lagrangian" and "Eulerian" perspectives mean in this context?
    - Can you please elaborate on lines 209-211? Why do we have to "learn strictly more than the velocity field"?
    - Regarding line 248: Can you explicitly show the correspondence to shortcut models?
    - Regarding line 262: How do you do the smooth conversion from diagonal training to self-distillation?
    - Regarding lines 271-272: What do you mean by "use the instantaneous parameters as the teacher"? And can you elaborate on how EMA is implemented here?
    -  Why is the flow map a function of two times?
    - Regarding line 598: How are the hyperparameter sweeps done?
    - Can you clarify: Is the loss in Eq. 12 enforced for all models (LSD, ESD, PSD-U, PSD-M)? Please clarify exactly which losses are used for these four models.

**Ethical Concerns:**

["NO or VERY MINOR ethics concerns only"]

**Final Justification:**

The authors provided reasonable explanations to my questions that bolster the contribution.

**Limitations:**

I felt that the technical limitations could use more elaboration – see above comments.

**Paper Formatting Concerns:**

No concerns.

**Quality:**

3

**Strengths And Weaknesses:**

Strengths:
1. Further elucidates the theoretical relation between the flow map and the vector field in the context of flow matching and diffusion models.
2. Provides some insightful practical details, e.g., about gradient norms under each of the proposed models.
3. Achieves positive results on CIFAR-10.

Weaknesses
1. The trend observed in Checkers (Table 1.) – PSD better than LSD for more steps – seems to be opposite of the trend observed for CIFAR-10 – PSD better than LSD for fewer steps. More justification or elaboration would help to make conclusions more concrete.
2. No means and standard deviations are given for Tables 1 and 2 over multiple runs. This would make the results more robust and perhaps better characterise the trends mentioned in the previous point.
3. Assumes a specific parameterisation of the flow map (Euler step). It would help to justify this more and elaborate on related limitations.
4. Assumes training only on the upper triangle – time pairs in the noise-to-data direction. No discussion, as far as I can tell, on training in the full unit square of time. Also, no discussion on the choice of a two-time flow map rather than one-time.
5. On first read, the notation is sometimes a bit hard to keep track of, and the correspondence between the loss functions and the tested models could be clearer.
6. Discussions of some ideas in the paper – e.g., Multiple Models (line 209) and  General Representations (line 218) – give the impression that they have been included in experiments (since they are included in the section "Algorithmic aspects". I would suggest making it more explicit which discussions pertain to ideas for future work and which pertain to experiments implemented in the paper.

---

> ### Author Rebuttal · Authors · 2025-07-31
>
> We thank the reviewer for their thorough and thoughtful comments on our work. We provide aggregate responses to their mentioned weaknesses and the questions related to them below. Due to the 10k character limit, we had to remove some, but have answers to all and are happy to provide them in the discussion period. If satisfactory, we would greatly appreciate an improved score.
>
> **CIFAR-10 vs. Checkerboard.** Our goal in this work is to introduce a systematic framework for training consistency models. We hypothesize that the optimal training scheme depends on the network architecture, the dataset, and the modality, which is why we lay out a family of methods rather than one. Our results support this hypothesis, with low-dimensional datasets exhibiting distinct trends from higher-dimesional vision datasets. To make this more clear, we have added additional text to the discussion and introduction that explicitly describes these distinct trends. As described in our reply to Reviewer oUCA, we have also performed additional experiments on the CelebA-64 dataset, and these new results support our claims even further: LSD performs best on Celeb-A, even though it is higher dimensional than CIFAR, suggesting the performance is very task dependent.
>
> **Error bars.** In our experience, variance in quantitative results due to network initialization is small. This is supported by common practices for results in the field: we performed a literature review and found that none of the major papers on consistency models included error bars [1-5]. For this reason, during this rebuttal period, we have focused on the generation of additional results rather than on quantifying error for our existing results. Nevertheless, we are happy to include them for the camera-ready version.
>
> **Euler parameterization.** While the parameterization that we exploit is reminiscent of an Euler step, we emphasize that the ansatz $X_{s, t}(x) = x + (t-s)v_{s, t}(x)$ can be used without loss of generality and introduces no approximation error. To see this, we may observe that the function $v_{s, t}(x) = \frac{1}{t-s}(X_{s, t}(x) - x)$ is simply a time-dependent shift and rescaling of the flow map. We have amended the text to stress there is no limitation in using it, it just allows us to avoid using autograd to take a time derivative.
>
> We also stress our algorithmic formulation is general and allows for an arbitrary parameterization of the map, though this choice is natural for the reason just described.
>
> **Upper triangle.** We trained over the upper-triangle because it corresponds to the generative pathway. Our framework is identical over the entire unit square. Earlier work has demonstrated applications of this to style transfer and image editing [6], which could also be performed using our methodology. In the revision, we have updated the text with a remark describing this point.
>
> **Two-time Maps.** There are several reasons to train a two-time map rather than a single-time map $f_s(x) = X_{s, 1}(x)$, which corresponds to a consistency model.
>
> With a single-time map, there is no principled way to sample in multiple steps that increases accuracy. Consistency models alternate between adding noise and applying the map, which only has a small effect on quality [2,4,5]. This fails because adding noise jumps between trajectories of the probability flow ODE, and hence has no convergent limit. In our approach, we converge to an integral of the underlying flow:
>
> $$\lim\_{k\to\infty} \circ\_{i=1}^k \hat{X}\_{t\_{i-1}, t\_i}(x\_0) = x\_0 + \int\_0^1 \hat{v}\_{\tau,\tau}(x(\tau))d\tau, \qquad 0 = t_0 < t_1 < ... < t\_k = 1,$$
>
> which enables us to trade inference-time compute for sample quality. As shown by our numerical results, this is observed in practice, with a higher number of steps leading to better quantitative results on all problems.
>
> The one-time map is also more limited from a training perspective. Because $f_s$ has neither $t$ dependence nor a semigroup property, there is only an Eulerian representation and neither LSD nor PSD can be used.
>
> **Notation.** We agree that the notation in our submission was cumbersome. The major culprit was a typo that incorrectly used the spurious notation $\mathcal{L}\_{\mathsf{SSD}}$ in place of the objective $\mathcal{L}\_{\mathsf{PSD}}$. This typo clashed with the $\mathcal{L}\_{\mathsf{SD}}$ self-distillation objective. In the revision, we have corrected this error, and we propose the following notation:
>
> - $\mathcal{L}_{b}$: flow matching / interpolant objective.
> - $\mathcal{L}\_{\mathsf{D}}$: distillation objective, taken in practice to either be $\mathcal{L}\_{\mathsf{LSD}}$, $\mathcal{L}\_{\mathsf{ESD}}$, or $\mathcal{L}\_{\mathsf{PSD}}.$
> - $\mathcal{L}\_{\mathsf{SD}} = \mathcal{L}\_{b} + \mathcal{L}\_{\mathsf{D}}$: self-distillation objective instantiated by choice of $\mathcal{L}\_{\mathsf{D}}$.
>
> We have added a section to the appendix that provides a glossary on notation, as well as provides the recommended form of each objective function, including the use of $\mathcal{L}_b$.
>
> **Algorithmic aspects.** We agree that the presentation of this section may have been confusing because the two future directions came first. We have re-ordered it so that the future directions are presented *last*, and we have added text emphasizing that they were not explored in the present work.
>
> **$\eta$-sampling.** We have added the following discussion to the appendix, which quantifies each method's cost.
>
> Evaluating the interpolant loss $\mathcal{L}_b$ on a single sample requires a single neural network evaluation $b_t(I_t)$, leading to $B$ network evaluations on a batch.
>
> The LSD objective requires a single partial derivative evaluation $\partial_t X_{s, t}(I_s)$ and two network evaluations -- one for $v_{t, t}$ and one for $X_{s, t}$ -- per sample. The time derivative is a constant factor $C \approx 1.5$ more than a forward pass. The LSD objective thus requires $(2 + C)B$ network evaluations. Adding the diagonal and off-diagonal parts, we find a complexity of $((1-\eta)(2 + C) + \eta)B$ for the full self-distillation objective.
>
> The PSD objective requires three neural network evaluations, so that its expense is $3B$. Combining this with the diagonal component, we have $(3(1-\eta) + \eta)B$ network evaluations.
>
> The ESD objective requires a partial derivative evaluation $\partial_s X_{s, t}(I_s)$, a neural network evaluation $v_{s, s}(I_s)$, and a Jacobian-vector product $\nabla X_{s, t}(I_s)v_{s, s}(I_s)$, leading to a complexity of $(2C + 1)B$. Adding the diagonal component, we find $((2C+1)(1-\eta) + \eta)B$.
>
> The factor $\eta$ can be chosen based on the available computational budget to systematically trade off the relative amount of direct training and distillation per gradient step.
>
> **Training Cost.** With the addition of our new results on the CelebA-64 dataset, we feel that our text answers this question to a complete level. On the checkerboard, we fixed the total number of iterations, highlighting that when allowing for differences in computational expense we obtain better results with the cheaper LSD and PSD methods. On the image datasets CIFAR-10 and Celeb-A, we fix the total training time, allowing for different numbers of iterations. We find a similar pattern, where LSD and PSD outperform ESD. To make this more clear, we have added further empirical detail both to the main text and appendix.
>
> **Multiple models.** We have performed these ablations on MNIST and find high-quality visual results in both cases. In particular, we tried the higher-order / multiple-model representation, as well as the more general representation $X_{s, t}(x) = x + \phi_{s, t}(x)$. In the camera-ready version, we will include these results in the appendix.
>
> **KL Computation.** We generate $10^6$ samples from the learned models and compute approximate densities via histogram. Because we have access to the analytical functional form of the checkerboard density, we integrate the definition of the KL divergence $\mathsf{KL}(\rho^* || \hat{\rho}) = \int \log(\frac{\rho^*(x)}{\hat{\rho}(x)})\rho^\*(x)dx$ via quadrature. In the camera-ready version, we will include error bars. We'll include details on this KL computation in the appendix.
>
> **Fairness of comparison.** We would like to stress that our results show that no single objective is universally best; performance depends on the network and task. The Lagrangian and Eulerian losses more strongly enforce the dynamical equations for the map, but they require gradients through space and time derivatives, which can be high variance for complex models like UNets. On the checkerboard, where the network is a simple MLP, these stronger objectives excel. We have clarified this in the revision and proposed exploring architectures better suited to such losses as a future direction.
>
> **Shortcut Models.** We have included a new section in the appendix that gives the details of this mapping. Let $\Delta = t-s$. If we re-write $$X_{s, t}(x) = X_{s, s+\Delta}(x) = x + \Delta \, v_{s, s+\Delta}(x),$$ use the PSD-M objective with PSD scaling, and place a stopgrad operator on the teacher, our loss function becomes $$\mathcal{L}\_{\mathsf{PSD}}(\hat{v}) = \int\_0^1 \int\_0^t \mathbb{E}\left[|\hat{v}\_{s, s+\Delta}(I_s) - \frac{1}{2}\mathsf{sg}(\hat{v}\_{s, s+\Delta/2}(I_s) + \hat{v}\_{s+\Delta/2, s+\Delta}(\hat{X}\_{s, s+\Delta/2}(I\_s)))|^2\right]dsdt.$$ This is precisely the shortcut model objective function.
>
>
> ## References
> [1] Geng et al, “Mean Flows for One-Step Generative Modeling.”
>
> [2] Song et al, “Consistency Models.”
>
> [3] Kim et al,  “Consistency Trajectory Models: Learning Probability Flow ODE Trajectory of Diffusion.”
>
> [4] Lu et al, “Simplifying, Stabilizing and Scaling Continuous-Time Consistency Models.”
>
> [5] Song et al, “Improved Techniques for Training Consistency Models.”
>
> [6] Boffi et al, "Flow Map Matching with Stochastic Interpolants: A Mathematical Framework for Consistency Models."

---

> > ### Comment · Reviewer_cfDL · 2025-08-06
> > **Thanks for your rebuttal**
> >
> > Thanks for your rebuttal. Most of the responses seem fair. I have a few remaining questions/comments.
> >
> > **Euler parameterization**
> > I agree that the chosen flow map is just a shift of the flow map. But could there be any benefit to using, e.g., an RK45 parameterization?
> >
> > **Upper triangle**
> > I understand that the upper triangle corresponds to the generative pathway. But, in theory, since the flow map should be able to map to any time, would there be any benefit to also training on the noising pathway? Or is there some inductive bias that makes training on only the upper triangle identical to training on both the upper and the lower triangles?
> >
> > **Different objectives**
> > Is there any evidence to show that some objectives work better for some datasets?
> >
> > Thanks.

---

> > > ### Author Response · Authors · 2025-08-06
> > >
> > > Many thanks for your reply! We answer your additional questions below -- please let us know if there is any further clarification we can provide.
> > >
> > > > I agree that the chosen flow map is just a shift of the flow map. But could there be any benefit to using, e.g., an RK45 parameterization?
> > >
> > > We would first like to emphasize that from a representational perspective our parameterization is unrestricted. This means that there is no fundamental theoretical benefit to using another functional form; the gains would purely be empirical.
> > >
> > > Given this observation, we would like to emphasize that our aim here is to reduce the computational complexity of generation, which requires reducing the number of network evaluations. If we use a parameterization inspired by an integration scheme such as RK45, a single evaluation of the flow map would necessarily require 4-5 network evaluations, which reduces the efficiency significantly. While it may be possible to get away with fewer "steps" of the flow map, this would be a misleading metric, as each step would be similar to a few-step integration of an ODE and would be much more expensive.
> > >
> > > By learning the flow map in our functional form, we essentially learn the integration scheme directly, rather than build it into the architecture. This allows us to use more network evaluations to improve the quality if we desire, but also enables us to use as few as one, giving us more granular control over the complexity of generation.
> > >
> > > The main advantage of the "Euler" parameterization is then to provide a derivative-free way to evaluate $\lim_{s\to t}\partial_t X_{s, t}(x).$ Another single-step parameterization that is not Euler-like is $X_{s, t}(x) = x + \phi_{s, t}(x)$, but this form would require computation of $\lim_{s\to t}\partial_t \phi_{s, t}(x)$, which is more expensive than our approach.
> > >
> > > > I understand that the upper triangle corresponds to the generative pathway. But, in theory, since the flow map should be able to map to any time, would there be any benefit to also training on the noising pathway? Or is there some inductive bias that makes training on only the upper triangle identical to training on both the upper and the lower triangles?
> > >
> > > This is a fantastic question that we are more than happy to investigate for the camera-ready version. Our hypothesis is that there will not be a benefit. We believe this to be the case because the data will be the same even when training over the entire unit square. The only change is the role of $s$ and $t$, but the interpolant $I_t$ is already used for all $t \in [0,1]$. In this sense, when training over the entire unit square, we will use the same data but will ask the network to be able to accomplish an additional task (inversion), which requires greater capacity. Nevertheless, we believe inversion is an interesting usage of flow maps and will test our hypothesis for the final version.
> > >
> > > > Is there any evidence to show that some objectives work better for some datasets?
> > >
> > > Thanks for asking this important question, which is central to our conclusions. We would like to politely emphasize that all of our results show that some objectives work better for some datasets (and architectures). On the checker, LSD and ESD perform better than PSD for few steps, but PSD performs the best for more steps. On both image datasets CIFAR-10 and CelebA-64, ESD is unstable, while it works well on the checker. On CIFAR-10, PSD outperforms LSD across all tested number of sampling steps. On CelebA-64, LSD outperforms PSD at all tested number of sampling steps, completely flipping the trend. In this way, we believe that our numerical experiments clearly demonstrate the interdependence between the learning objective, the dataset, and the network.
> > >
> > > Please let us know if and how we can provide any further clarification.

---

> > > > ### Comment · Reviewer_cfDL · 2025-08-07
> > > > **Thanks for your response.**
> > > >
> > > > Thanks for the responses to my points. What I meant regarding the interdependence between the learning objective, the dataset, and the network is: what is the justification for interdependencies? E.g., why do LSD and ESD perform better than PSD for a few steps? And why does PSD perform the best for more steps? And etc?

---

> > > > > ### Author Response · Authors · 2025-08-08
> > > > >
> > > > > Thank you for your clarification on this important question. We would first like to emphasize that the precise trend mentioned is true on the checkerboard dataset, but is not true for the image datasets CIFAR-10 and CelebA-64. For this reason, we would like to express caution in providing explicit justification for trends observed on specific datasets. Nevertheless, below we provide some high-level justification as to *why* we expect the emergence of different behavior.
> > > > >
> > > > > At an intuitive level, all loss functions are “equivalent” in the sense that the global minimizer of each is guaranteed to be the target flow map. Despite this, precisely how they minimize the “discrepancy” between the current estimate $\hat{X}$ and the target $X$ is very different, which leads to different sources of error in the learned map.
> > > > >
> > > > > The ESD objective minimizes the square residual of a partial differential equation that must be satisfied globally by the flow map. This equation defines a strong pointwise relationship between the spatial and time derivatives of the map. These derivatives, and in particular the spatial derivative, are known to be poorly behaved for complex network architectures such as UNets. Moreover, they are known to be particularly poorly behaved for optimization (i.e., backpropagating through the spatial derivative is often unstable). For this reason, we believe the worse performance and instability of ESD to be fundamentally an optimization-related difficulty. This is supported by recent efforts such as Align Your Flow and Mean Flow, which are based on objectives similar to ESD but which leverage the stopgrad operator to stabilize the optimization.
> > > > >
> > > > > The LSD objective imposes that the flow map locally satisfies the original probability flow ODE, in the sense that tangents to all curves defined by the map should be given by the velocity field $v_{t, t}$. It eliminates the dependence on a spatial derivative, making it more amenable to optimization over complex neural networks, which explains its improved performance compared to ESD. One reason we may expect it to perform better than PSD for a small number of steps is because it directly imposes a condition that must be satisfied globally by the optimal flow map for all $(s, t)$ in terms of the flow $v_{t, t}$.
> > > > >
> > > > > By contrast, the PSD objective minimizes the square residual for the semigroup / composition property $X_{s, t} = X_{\tau, t}\circ X_{s, \tau}$. This means that the ability to take large steps is “bootstrapped” from the ability to take small steps via composition. Infinitesimal steps are learned through the flow $\hat{v}\_{t, t}$, which are converted into steps of finite size through the semigroup condition. While this works in the limit, in general we expect there to be compounding errors: because the flow map is not learned perfectly for small steps, and because these small steps are imperfectly converted into larger steps, we expect the error to be largest for the few-step map. Notably, this precise issue arises when trying to prove Wasserstein bounds for the PSD algorithm, and we believe it to be fundamental. LSD does not require bootstrapping from small steps to larger steps (it only requires the implicit signal from $v\_{t, t}$), and hence performs better for a small number of steps.
> > > > >
> > > > > We will include a paragraph in the discussion section describing these important considerations in the camera-ready version.

---

### Official Review · Reviewer_oUCA · 2025-07-17

**Clarity:** 4
**Significance:** 3
**Originality:** 3
**Rating:** 5
**Confidence:** 4

**Summary:**

This paper introduces a unifying framework that encompasses a wide range of existing distillation schemes.  By analyzing how different conditions imposed by their characterisation of the flow map, alongside the insight that the velocity field corresponds to the instantaneous rate of change of the flow map, the authors reveal connections between established distillation algorithms. Building on this perspective, the paper proposes a novel self-distillation algorithm to train flow maps. The authors prove that their training scheme and the associated loss functions are well-posed—meaning each admits a unique solution. Additionally, they provide theoretical guarantees in the form of bounds on the 2-Wasserstein errors from the loss values for 2 of the loss function proposed in their paper. Empirical evaluations demonstrate that the proposed training approach and loss functions achieve promising results for unconditional generation, validated on both synthetic datasets (such as checkerboard patterns) and the CIFAR-10 benchmark.

**Questions:**

1. Conduct and report systematic hyperparameter sweeps for key variables such as learning rate, model depth, regularization parameters, and optimizer hyperparamters and choice. Analyze how sensitive the training dynamics and performance are to these settings. Highlight scenarios where the method exhibits instability or performance drops.
2. Experiment with alternative flow map architectures—varying network depths, widths, or activation functions—to identify which design choices most affect robustness. Discuss not only successes but also observed failure modes linked to specific architectures.
3. Evaluate the method using more challenging datasets, such as high-resolution or multi-modal benchmarks (e.g., CelebA) or even datasets for discrete random variablaes. This will help uncover common failure modes that may not be apparent with simpler data.

**Ethical Concerns:**

["NO or VERY MINOR ethics concerns only"]

**Final Justification:**

The authors have done more experiments and added more theoretical results to strengthen their claims as well as clairfy them. This justifies the changes in score.

**Limitations:**

Yes

**Paper Formatting Concerns:**

None.

**Quality:**

4

**Strengths And Weaknesses:**

Strengths
1. The paper presents a comprehensive framework that unifies a wide range of distillation methods, offering a cohesive perspective that deepens the understanding of existing approaches and an approach to how new distillation algorithms can be systematically derived.
2. The authors provide theoretical guarantees for their framework, which also offer insights into its relationship with related algorithms.
3. The framework opens up interesting possibilities, such as incorporating multiple models and exploring additional extensions.

Weakness
1.  While the results on the 2D checkerboard dataset are promising, the empirical evaluation is restricted to relatively simple datasets. The case for the method’s effectiveness would be much stronger if it were demonstrated on more complex, high-resolution, and multi-modal datasets, such as CelebA
2. The descriptions of the algorithmic aspects—particularly loss reweighting, as well as temporal and PSD sampling—are at times difficult to follow; the paper would benefit from clearer exposition or additional explanation in these sections.
3. The discussion around hyperparameter sweeps and their impact when training flow maps for CIFAR-10 in the appendix could be expanded to provide more useful insights for the reader.

---

> ### Author Rebuttal · Authors · 2025-07-31
>
> We thank the reviewer for their careful reading and evaluation of our work. Below, we address the main questions and weaknesses raised. If our arguments are found convincing, we would greatly appreciate the reviewer raising their score. We would equally appreciate further suggestions for us to improve our work, and will happily provide further clarification on any aspects of our paper or our answers.
>
> ## Weaknesses
> *While the results on the 2D checkerboard dataset are promising, the empirical evaluation is restricted to relatively simple datasets. The case for the method’s effectiveness would be much stronger if it were demonstrated on more complex, high-resolution, and multi-modal datasets, such as CelebA.*
>
> We agree with the reviewer that the initial evaluation was limited, and we thank them for drawing our attention to this point. Since receiving the reviews, we have launched a round of experiments on CelebA-64. In these experiments, we first anneal by training only on the diagonal $s=t$ for 24 hours on 4x H100s with a batch size of 256. Sampling directly with the learned $\hat{v}\_{t, t}$ achieves an FID of 4.3 using 64 steps of the Heun sampler. We leverage an architecture based on the small EDM2 configuration for ImageNet64 [1], using $128$ channels for the estimated log-variance weight factor $w_{s, t}$, $128$ channels at the first layer of the UNet hierarchy, a four-layer hierarchy with channel multipliers in the pattern $(1, 2, 3, 4)$, three residual blocks per layer, positional encoding, attention at the $8\times 8$ and $16\times 16$ resolutions, and zero dropout. We then expand the range of $|s-t|$ linearly away from the diagonal for $10,000$ gradient steps and train for an additional 24 hours using the full self-distillation methodology. We study the performance as a function of the algorithm and the number of sampling steps.
>
> Our empirical FID scores are shown in the table below. What we find confirms our results on CIFAR-10, again highlighting dataset-dependence and differences between stylized low-dimensional examples and images. Our results demonstrate that LSD and PSD outperform ESD, which is again unstable. In this case, LSD performs the best at all numbers of sampling steps.
>
> | Method | 1 step | 2 step | 4 step | 8 step | 16 step |
> |--------|--------|--------|--------|--------|---------|
> |   LSD    | **15.77**       | **8.53**       |  **4.70**      |  **3.20**      | **2.66**        |
> |   PSD-M     |  17.24      |   9.34     |   5.77     |  3.97      |  3.25       |
> |   PSD-U     |  17.94      |  9.19     |   5.55     |   4.11    |   3.55      |
>
> Given time constraints and an inability to upload new visual results with this year's rebuttal format, we chose to focus solely on Celeb-A. For the camera ready version, we also plan to launch a round of experiments on the LSUN-Bedrooms dataset to further validate our findings, and are currently running some experiments with the SiT architecture on ImageNet-256 [2].
>
> *The descriptions of the algorithmic aspects—particularly loss reweighting, as well as temporal and PSD sampling—are at times difficult to follow; the paper would benefit from clearer exposition or additional explanation in these sections.*
>
> We thank the reviewer for bringing our attention to this point. We have significantly re-worked this section to improve the clarity of presentation. In particular, we have:
>
> - Clarified how to make use of multiple models. To this end, we have added a section to the appendix explicitly providing the loss $\mathcal{L}(\hat{b}, \hat{X}) = \mathcal{L}_b(\hat{b}) + \mathcal{L}_D(\hat{X}; \hat{b})$ that simultaneously trains a flow model $\hat{b}$ and distills it into a teacher.
> - Clarified our representation. In particular, we have highlighted that our functional form $X_{s, t}(x) = x + (t-s)v_{s, t}(x)$ is not an approximation: it can be used without loss of generality, and was chosen for computational efficiency, because it leads to the expression $\lim_{s\to t}\partial_t X_{s, t}(x) = v_{t, t}(x)$. We have also explained more clearly how taking additional terms allows us to split the role of the velocity field $b_t$ and the "residual" term $\psi_{s, t}$, for example in the representation $X_{s, t}(x) = x + (t-s)b_t(x) + \frac12(t-s)^2\psi_{s, t}(x)$.
> - We have updated the description of the learned loss weight $w_{s, t}$ to clarify its role. Originally introduced in the EDM2 paper [1], this term learns to weight different times in the loss equally, leveraging insights from multitask learning. We have also added a section to the appendix, generalizing the derivation from [1] to the two-time setting, highlighting that the minimum over $w_{s, t}$ normalizes the variance of the loss between the different time points $(s, t)$.
> - We have clarified the role of the sampling distribution $p_{s, t}$ and how it differs from the weight $w_{s, t}$. The sampling distribution $p_{s, t}$ defines how we "select" time points for the Monte-Carlo approximation of the loss, while the weight $w_{s, t}$ defines how much each such pair contributes to the loss. We have also significantly simplified our discussion of how we select the sampling distribution $p_{s, t}$, which we agree was cumbersome in the original submission. We have clarified the differences between the uniform distribution on the diagonal and uniform on the off-diagonal, as well as the role of the $\eta$ parameter in tuning the computational effort, and moved many of the technical details to the appendix. For further computational detils on the choice of $\eta$, please see our response to Reviewer cfDL.
> - We have extended and strengthened the discussion of PSD sampling to make this important design decision more clear. In PSD, the flow map model $\hat{X}$ learns to make a jump from time $s$ to time $t$ equal to the two-step jump from $s$ to $u$ and then from $u$ to $t$. The sampling distribution $p_u$ then defines how this intermediate time $u$ is selected. A natural way to parameterize this time $u$ is to write it as a convex combination of $s$ and $t$, $u = \gamma s + (1 - \gamma)t$ for $\gamma \in [0, 1]$. This parameterization is useful because $\gamma$ is always on the same scale independent of the selected values of $s$ and $t$. In the original version, we show in the appendix that re-writing the objective function in terms of $\gamma$ preconditions the loss by placing terms on the same scale. In the revision, we have updated this discussion, highlighting more explicitly how this preconditioning removes terms such as $t-u$ that can be on a different scale depending on the values of $(s, u, t)$.
>
> *The discussion around hyperparameter sweeps and their impact when training flow maps for CIFAR-10 in the appendix could be expanded to provide more useful insights for the reader.*
>
> We agree with the reviewer that the original discussion on experimental details for CIFAR-10 was somewhat sparse, which we have significantly expanded as a result. We have clarified the specific network architecture by providing a reference to Configuration G of [1], along with exact definitions of each architectural hyperparameter. In addition, we have added further details on the annealing schemes -- in particular, the precise configuration of loss functions we use over the training trajectory, how the anealing is performed, how many GPUs were required, and the timing information. We have added an analogous section for our new results on the Celeb-A dataset.
>
> ### References
>
> [1] Karras, Tero, Miika Aittala, Jaakko Lehtinen, Janne Hellsten, Timo Aila, and Samuli Laine. “Analyzing and Improving the Training Dynamics of Diffusion Models.” arXiv:2312.02696.
>
> [2] Ma, Nanye, Mark Goldstein, Michael S. Albergo, Nicholas M. Boffi, Eric Vanden-Eijnden, and Saining Xie. SiT: Exploring Flow and Diffusion-Based Generative Models with Scalable Interpolant Transformers. arXiv:2401.08740.

---

> > ### Comment · Reviewer_oUCA · 2025-08-05
> >
> > The clarifications are useful and I am happy to see more details as well as new results as well.

---

> > > ### Author Response · Authors · 2025-08-05
> > >
> > > Many thanks for your reply. Please let us know if there is any further clarification that we can provide.

---

### Comment · Area_Chair_f55u · 2025-08-06
**Baselines**

Dear authors,

everybody is appreciating the theoretical and methodologically unifying contributions of the paper, which is great!

However, I would like to expand on a remark by reviewer 2Efm regarding baselines:

Existing work on training few-step models from scratch outperform this approach significantly. For example, Easy Consistency Tuning achieves an 1-step FID of 3.60 and 2-step FID of 2.11 without a pretrained model (https://arxiv.org/pdf/2406.14548, table 1). The present method achieves 14.13 respectively 8.59.

I agree with the reviewers that not every paper has to put forward a new SOTA, but why do the results fall short of the baselines in that table?

Many thanks!

-- AC

---

> ### Author Response · Authors · 2025-08-08
>
> Dear Area Chair,
>
> Thank you for your engagement with our work, and for pointing out the gap in baseline performance. We would like to emphasize that this gap is primarily computational in nature: our aim here was not to shoot for near-SoTA performance, but to compare performance of our different methods on an equal footing. For this reason, we ran each experiment for a fixed amount of time -- usually 24-36 hours on a few GPUs -- and compared performance between each algorithm with identical hyperparameters. We made little effort to optimize the hyperparameters individually for each algorithm, which would take significantly more computational power than we have access to.
>
> Obtaining the higher-performing results seen in some earlier papers on consistency models requires an immense amount of computational effort. For example, as quoted in [1], it can take as much as a week using 8 GPUs even on small datasets such as CIFAR-10. We are confident that our approach will perform just as well when scaled to this degree of computational effort.
>
> Outside of compute, we would also like to emphasize that we use several simple algorithmic choices -- such as uniform sampling of time -- that could be tuned and engineered to maximize performance. Other works spend significantly more effort on this engineering component, while our contribution is more conceptual in nature, and focuses on various choices of loss functions that arise from distinct mathematical characterizations. See, for example, the “tangent warmup”, time sampling, loss weighting, and curriculum techniques introduced in [2], which we expect could be adapted to our setting in follow-up works.
>
> We would also like to mention that Easy Consistency Tuning [1] requires a pre-trained model, as stated in their abstract and in their experimental results. While the “tuning” or “post-training” phase can be performed quickly in some cases, they always begin with a near-SoTA pre-trained model, while we train all of our approaches from scratch under identical conditions for an apples-to-apples comparison.
>
> Last, we would like to emphasize that we do not use any FID “optimization” techniques such as additional denoising steps on the generated images, which often improve performance significantly.
>
> Please let us know if there is any additional information that we may provide about our manuscript. We remain at your disposal to clarify further.
>
> **References**
>
> [1] Consistency models made easy. Geng et. al. (2024).
>
> [2] Simplifying, Stabilizing and Scaling Continuous-Time Consistency Models. Lu et. al. (2024).

---

> > ### Comment · Area_Chair_f55u · 2025-08-08
> >
> > Dear authors,
> >
> > Thanks for the explanations. I will ask the reviewers to consider both my concern and your answer in their updated review.
> >
> > Best regards,
> >
> > AC

---

### Comment · Area_Chair_f55u · 2025-08-08
**Please update Rating and Final Justification**

Dear reviewers,

it is time to finalize your reviews if you have not done so.

Please take discussion with other reviewers into account and update your Rating and Final Justification until Wed, 13th. The justification should list which weaknesses you still think apply. If you have not posted a final answer to the authors, you can usually just copy over your final justification.

**Please also take the concern I raised about competing baselines into account. The gist is that the proposed method is worse than baselines in terms of FID. The authors argue that it's fine given that they use significantly less compute on training and tuning their method.**

If anything is unclear or you want to discuss/add a point, drop a comment below.

Thanks!

AC

---

### Note · Authors · 2025-08-15

Dear Reviewers and AC,

We would first like to thank both the reviewers and the AC for their thoughtful comments, critiques, and suggestions on the manuscript. We believe that the discussion period has led to a greatly improved paper by strengthening both the presentation of our formalism and our empirical results. We feel that our updated results on Celeb-A further highlight the tradeoffs between the various training schemes for the flow map, emphasizing how the choice of optimal training algorithm is intimately connected to both the architecture and the dataset. We also feel that our updated main text and appendices further clarify the design space of flow map models, emphasizing how our approach connects to the various algorithms that have been introduced in the literature and the precise training schemes and hyperparameter tuning we used to arrive at our results.

As a final remark, we would like to stress that the purpose of our paper is to provide a complete picture of how the equations that characterize the flow map can be used to design direct training schemes, as well as how these equations reduce to the majority of training paradigms that have appeared in prior work in particular cases. Using this picture, we are able to compare their performance on problems in both low and high dimension based on a variety of factors, such as the presence of spatial and temporal derivatives in the loss functions, the effect of stop-gradients, and the problem dimensionality.

We would like to explicitly emphasize that our aims are not to try to produce state-of-the-art FIDs. Our experiments use a limited but equal amount of compute so as to directly compare the methods, rather than to choose one and optimize for FID performance. Moreover, the numbers the AC reports for related work stem from initializing the flow map from a pre-trained diffusion model, whereas we are directly after the question: how does the purview of direct training schemes for flow maps compare on specific problems?

We hope that you can take these perspectives into consideration as you discuss and make a final decision. Thanks again for your insights and feedback that has helped improve our submission.

Best wishes,

The Authors

---

### Decision · Program_Chairs · 2025-09-17

**Decision:**

Accept (poster)

**Comment:**

The authors propose a few-step generative modeling framework that unifies a range of existing distillation methods.

Strengths:

- The paper draws a useful design space and theoretical characterization of large-step flow maps.
- Empirically, the proposed methods provide positive results.

Weaknesses:

- The original paper only provided results on smaller datasets (reviewers oUCA, 2Efm, ysYP, 1nz6). The authors added additional experiments on CelebA, again in the form of an ablation of methods as their results on CIFAR-10.
- The proposed methods falls short of state-of-the-art methods (AC, 2Efm). The authors attribute this to the extensive compute requirements for fully training such models.

The reviewers unanimously highlight the foundational work presented in the theoretical framework, warranting acceptance. With a more extensive empirical evaluation, this could easily have been a spotlight or oral paper. The authors promised to add LSUN-Bedrooms and ImageNet-256 experiments.